# Urban-rural disparities in institutional delivery among women in East Africa: A decomposition analysis

Reta Dewau[1]*, Dessie Abebaw Angaw[2], Getahun Molla Kassa[2], Baye Dagnew[3], Yigizie Yeshaw[3], Amare Muche[1], Dejen Getaneh Feleke[4], Eshetie Molla[5], Enyew Dagnew Yehuala[6], Sisay Eshete Tadesse[7], Melaku Yalew[8], Zinabu Fentaw[1], Ahmed Hussien Asfaw[7], Assefa Andargie[1], Muluken Genetu Chanie[9], Wolde Melese Ayele[1], Anissa Mohammed Hassen[1], Yitayish Damtie[8], Foziya Mohammed Hussein[7], Zinet Abegaz Asfaw[8], Elsabeth Addisu[8], Bezawit Adane[1], Fanos Yeshanew Ayele[7], Bereket Kefale[8], Aregash Abebayehu Zerga[7], Tefera Chane Mekonnen[7], Mogesie Necho[10], Oumer Abdulkadir Ebrahim[11], Metadel Adane[12], Tadesse Awoke Ayele[2]

1 Department of Epidemiology and Biostatistics, School of Public Health, College of Medicine and Health Sciences, Wollo University, Dessie, Ethiopia, 2 Department of Epidemiology and Biostatistics, Institute of Public Health, College of Medicine and Health Sciences, University of Gondar, Gondar, Ethiopia, 3 Department of Physiology, School of Medicine, College of Medicine and Health Sciences, University of Gondar, Gondar, Ethiopia, 4 Department of Neonatal Nursing, College of Medicine and Health Sciences, Debretabor University, Debretabor, Ethiopia, 5 Department of Public Health, College of Medicine and Health Sciences, Debretabor University, Debretabor, Ethiopia, 6 Department of Midwifery College of Medicine and Health Sciences, Debretabor University, Debretabor, Ethiopia, 7 Department of Public Health Nutrition, School of Public Health, College of Medicine and Health Sciences, Wollo University, Dessie, Ethiopia, 8 Department of Reproductive and Family Health, School of Public Health, College of Medicine and Health Sciences, Wollo University, Dessie, Ethiopia, 9 Department of Health System and Policy, School of Public Health, College of Medicine and Health Sciences, Wollo University, Dessie, Ethiopia, 10 Department of Psychiatry, College of Medicine and Health Sciences, Wollo University, Dessie, Ethiopia, 11 Department of Public Health, College Health Science, Samara University, Assayta, Ethiopian, 12 Department of Environmental Health, College of Medicine and Health Sciences, Wollo University, Dessie, Ethiopia

* retadewau30@gmail.com

**Data Availability Statement:** The data used in this study are third party data from Measure DHS (http://www.dhsprogram.com) and can be

## Abstract

### Background

Though institutional delivery plays a significant role in maternal and child health, there is substantial evidence that the majority of rural women have lower health facility delivery than urban women. So, identifying the drivers of these disparities will help policy-makers and pro-grammers with the reduction of maternal and child death.

### Methods

The study used the data on a nationwide representative sample from the most recent rounds of the Demographic and Health Survey (DHS) of four East African countries. A Blinder-Oaxaca decomposition analysis and its extensions was conducted to see the urban-rural dif-ferences in institutional delivery into two components: one that is explained by residence dif-ference in the level of the determinants (covariate effects), and the other components was explained by differences in the effect of the covariates on the outcome (coefficient effects).

accessed following the protocol outlined in the Methods section.

**Funding:** The authors received no specific funding for this work.

**Competing interests:** The authors have declared that no competing interests exist.

**Abbreviations:** ANC, Antenatal care; EDHS, Ethiopian Demographic and Health Survey; ES, Enumeration Areas; HF, Health Facility.

## Results

The findings showed that institutional delivery rates were 21.00% in Ethiopia, 62.61% in Kenya, 65.29% in Tanzania and 74.64% in Uganda. The urban-rural difference in institutional delivery was higher in the case of Ethiopia (61%), Kenya (32%) and Tanzania (30.3%), while the gap was relatively lower in the case of Uganda (19.2%). Findings of the Blinder-Oaxaca decomposition and its extension showed that the covariate effect was dominant in all study countries. The results were robust to the different decomposition weighting schemes. The frequency of antenatal care, wealth and parity inequality between urban and rural households explains most of the institutional delivery gap.

## Conclusions

The urban-rural institutional delivery disparities were high in study countries. By identifying the underlying factors behind the urban-rural institutional birth disparities, the findings of this study help in designing effective intervention measures targeted at reducing residential inequalities and improving population health outcomes. Future interventions to encourage institutional deliveries to rural women of these countries should therefore emphasize increasing rural women's income, access to health care facilities to increase the frequency of antenatal care utilization.

## Introduction

Despite progress in reducing maternal mortality worldwide and an ambitious plan has been set to reduce the maternal mortality ratio below 70 per 100,000 live births by the year 2030, there were an estimated 295,000maternal deaths due to pregnancy and child birth by the year 2017 [1, 2]. In 2016, maternal death was the second foremost cause of mortality for women of reproductive-age, following HIV/AIDS, and it was the primary cause among women aged 15–29 years [2]. Almost all maternal deaths (95%) were recorded in low and lower-middle-income countries, and nearly two-thirds (65%) happened in Africa [2].

In 2017, the highest number of maternal mortalities in Africa was estimated in Ethiopia and Tanzania next to Nigeria and the Democratic Republic of Congo [1]. The ratio of maternal deaths was 556 in Tanzania [3], 412 in Ethiopia [4], 362 in Kenya [5] and 336 in Uganda [6] per 100,000 live births.

Most of these passings were avoidable with legitimate clinical consideration and sterile settings during delivery and by lessening the dangers of inconveniences and disease to either the mother or the baby. That is why institutional delivery has been broadly promoted as a key approach to reducing maternal deaths [4, 5, 7–9].

However, according to the 2016 Ethiopian Demographic Health Survey (EDHS), the proportion of facility institutional deliveries attended in Ethiopia was still desperately lower compared to countries in sub-Saharan Africa (SSA) 47% [4]. Even though women have accessed cost-free maternal and newborn care services, the proportion of institutional deliveries were in Ethiopia 26% [4], Kenya 61% [5], in Tanzania 63% [3] and in Uganda 74% [6]. Yet the goal of institutional delivery is to save the lives of mothers and their newborns through skilled health professional assistance and quality care [2, 4].

Moreover, evidence also revealed that urban-rural significant disparities in health facility deliveries in developing countries [4, 5, 9]. According to the EDHS 2016 report, of the total

(26%) institutional deliveries, 79% of deliveries were in urban areas and only 21% of births in rural women [4]. In Kenya, urban institutional delivery was 82% and rural residents received 48.5% [5], in Tanzania it was 86.4% and rural residents received 53.7% [3] and in Uganda it was 87.8% in urban areas and 69.5% in rural areas [6]. It is well known that home delivery is particularly high in developing countries, especially among the poorest households, and rural area residents. There is considerable evidence that rural women gave home delivery compared to their urban counterparts [4, 5, 9, 10]. For instance, decomposition analysis unfolded that urban residences in India and Ghana positively contributed to the facility delivery inequalities [10, 11]. Another study conducted in Ethiopia [12–14], Guinea-Bissau [15], and Malawi [16] also showed similar findings. These urban-rural discrepancies were also well documented in different countries, DHS and WHO reports [2, 4].

From different literatures, various explanatory variables were significantly linked with facility-based delivery women education [10, 13, 14, 16–19] wealth index [10, 12, 17, 20], antenatal care [10, 13, 17, 19–21], husband's educational status [12, 18], average distance from health facility, birth order [12], mass media exposure [14, 18, 19, 21], maternal age [12, 14], marital status [16], parity [16, 22], age at first pregnancy [22], occupational status [22], types of pregnancy [12, 22, 23], and frequency of ANC Visit [22]. However, the decomposition analysis method helps to decompose inequalities in institutional deliveries in urban-rural settings. This difference was not rigorously explained using the decomposition method. Since the majority (64–80%) of Kenya, Tanzania, Uganda and Ethiopian population reside rurally [3–6] and their DHS data collection period proximity, we selected these countries for the analysis. Further progress in national maternal health outcomes cannot be realized, while not increasing facility delivery in rural areas. To develop effective strategies that minimize this urban-rural gap demands such, a greater understanding of the underlying factors contributing to these disparities in institutional deliveries between urban and rural areas has paramount importance. So, this study aimed to identify the sources of variation in institutional delivery between the urban and rural areas in East Africa by using a decomposition analysis method.

## Methods

### Source of data and population

This study used data from the most recent rounds of the Demographic and Health Survey (DHS) from Ethiopia, Kenya, Tanzania and Uganda conducted from 2014 to 2016 [3–6]. The DHS is an international survey conducted in 90 countries with the main objective of improving the collection, analysis, and dissemination of population, health, and nutrition data and to facilitate the use of these data for planning, policy-making and program management in developing countries. DHS collected data involving women 15–49 and men 15 to 59 as well as children under five. However, for the current study our analysis was restricted to only individual women's dataset [24]. The source population for this study was all women of reproductive age (15–49 years) in the study area. The sampled population was all reproductive-age women in each household in the enumeration area, and focused on their deliveries during the 5 years preceding the survey for each country.

### Sampling and sample size

Urban-rural based stratified two stage cluster sampling was employed for all countries. Stratification was done by region as well as place of residence for Ethiopia. In the first stage, clusters were selected based on the Probability Proportional to Size (PPS) method and in the second stage, equal numbers of households were selected using systematic random sampling [3–6]. From Ethiopia, Kenya, Tanzania and Uganda, 645, 1612, 608 and 697 clusters were sampled in the first stage respectively. For Ethiopia 28, Kenya 25, Tanzania 22 and Uganda, 30 households

per cluster were selected in the second stage. A total of 92548 households (18,008 in Ethiopia, 40,300 in Kenya, 13,360 in Tanzania and 20,880 in Uganda) were sampled. Overall, the household response rate was 99% for Kenya and 98% for Ethiopia, Tanzania and Uganda. From identified eligible women, the response rate for women interviewed was 95% for Ethiopia and 97% for the rest of the countries. The current study incorporates 643 clusters and 10,547 births from Ethiopia [4], 1,593 clusters and 20,840 births from Kenya [5], 608 clusters and 10,175 births from Tanzania [3] and 696 clusters and 15,154 births from Uganda [6].

## Study variable measurement

**Dependent variable.** Institutional delivery was the outcome of interest and was assessed using self-reported data on the place of delivery of all births that happened within 5 years of the dates of the surveys. For analysis, it was measured as a binary variable and was classified as "1" if a woman delivered in any healthcare facility and otherwise "0".

**Equity Stratifier variable.** Area of residency was the key independent variable and was a binary; classified as urban or rural.

**Explanatory variables.** Additional variables were extracted from the recent DHS surveys of four countries and they were treated as covariates in all analyses. These included factors that are known to be associated with institutional delivery such as; education level (ordinal categorical variable with categories 'no education', 'primary education', 'secondary education', and 'higher education'), wealth index (ordinal categorical variables with"poor","middle", and"rich"), and distance from a health facility (binary variable classified as one if the woman considered the distance to be a barrier to accessing care and zero otherwise), Husband's educational status (ordinal variable with "no education", "primary education", "secondary education", and "higher education"), occupational status (nominal categories with "not working", "agricultural" "government employed" or "manual worker") and media exposure (dummy variable with "exposed" or "not exposed"). Socio-demographic factors include marital status (categorical variable with "Not married," "married," or "widowed" and age in years at birth of a young child (continuous variable created from the difference between the year of child birth (b2) and the year of maternal birth (V010). marital status (categorical variable with "Not married", "married" or"widowed"). Antenatal care visits attended (categorical variable with "none", and "one and above"), and parity (ordinal categorical variable representing the number of children born to a woman "1", "2 to 4" and "5 and above "). These classification was made based on previous literatures [11, 12, 16, 17, 23, 25].

## Operational definition

*Media exposure.* Women are considered as regularly exposed to media if they have at least one of the media (newspaper, radio or television) at least once a week otherwise considered as have no access [4].

*Wealth index.* Based on the number and kinds of consumer goods they own, households are given scores. Principal component analysis was used to derive these scores. National wealth quintiles are compiled by assigning the household score to each usual household member, and then dividing the distribution into five equal categories, each comprising 20% of the population from the lowest poorest to richest for the current analysis we combine poorest and poorer as poor, richest and richer as rich and middle as it is [4].

## Statistical analyses

Data management and statistical analyses were conducted using STATA/MP 16.0 software. Means with standard deviation (SD) for normally distributed and median with Interquartile

Range (IQR) for skewed continuous variables and frequency with percentages for categorical variables were calculated to describe the characteristics of the study population by area of residency. Skewness and kurtosis test with normal quantile plot were used to assess normality. During analysis the survey design has been taken into account. The Pearson chi-square test was used to examine whether the place of delivery in urban and rural women's was statistically significant for categorical variables and independent t-test was used to compare the mean difference between urban and rural. To explain the urban-rural disparities in place of delivery among reproductive-age women, the Blinder-Oaxaca decomposition analysis was used. This technique decomposes the differences in an outcome for two groups can be explained by differences in the level or distribution of the determinants of the outcome (explained component/covariates effect), differences in the impact of these determinants on the outcome (unexplained component/coefficients effect), and/or the interaction of the two [26].

For example, if $Y_i$, was the outcome variable, and an independent variable X and we have two groups, urban and rural, then the place of delivery for the rural, and urban women are given as$\varepsilon$

$$Y_i^{rural} =_{\beta}{}^{rural}X_{i+\varepsilon}{}^{rural} \tag{1}$$

$$Y_i^{urban} =_{\beta}{}^{urban}X_{i+\varepsilon}{}^{urban} \tag{2}$$

Thus the urban-rural gap in the mean place of delivery ($Y^{urban} - Y^{rural}$), is given as

1. **Oaxaca decomposition**

$$Y^{urban} - Y^{rural} = (X^{urban} - X^{rural})\beta rural + (\beta^{urban} - \beta^{rural})X^{urban} + (X^{rural} - X^{urban})(\beta^{rural} - \beta^{urban}) \tag{3}$$

$$Y^{urban} - Y^{rural} =_{} \Delta X\beta^{urban} + \Delta\beta X^{rural} + \Delta X\Delta\beta \tag{4}$$

$$= E + C + CE \tag{5}$$

Where $\Delta X$ is the mean difference explanatory variables ($X^{urban}$—$X^{rural}$) and similarly $\Delta\beta = \beta^{urban} - \beta^{rural,}$

2. **Blinder decomposition**

$$Y^{urban} - Y^{rural} = (X^{rural} - X^{urban})\beta rural + (\beta^{rural} - \beta^{urban})X^{urban} + (X^{rural} - X^{urban})(\beta^{rural} - \beta^{urban}) \tag{6}$$

$$Y^{urban} - Y^{rural} = \Delta X\beta^{rural} + \Delta\beta X^{urban} + \Delta X\Delta\beta \tag{7}$$

$$= E + C + CE \tag{8}$$

So that the gap in average outcomes can be thought of as deriving from a gap in endowments (E), a gap in coefficients (C), and a gap arising from the interaction of endowments and coefficients (CE).

The Oaxaca [26] decomposition (4) utilizes the high group (urban women in this study) as the reference group, weighting contrasts in attributes by the coefficients of urban women and contrasts in coefficients by the covariates of rural women. The Blinder decomposition (7) on the other hand, utilizing the low group as the reference group (rural women in this study), and weighting contrasts in characteristics by the coefficients of the provincial women and differences in coefficients by the covariates of the urban women. The other methods used a weighted average of the two groups as weight. According to Reimers [27], the weighted mean should be

computed as 0.5 (equal weights for the two groups), whereas Cotton [28] assumes it ought to be the extents of the two groups in the sample. Since the result of the decomposition is delicate to the weighting technique utilized, this study ran an alternate decomposition for each method: Oaxaca, Blinder, Reimers, and Cotton. Regressions were performed for urban and rural women independently and afterward the assessed coefficients and covariates were utilized to compute the decompositions. Consistent results using the different weights were thought to represent robustness of the study outcome. Powers et al. [29] developed a recent extension of the Oaxaca-Blinder method for non-linear dependent variables, mvdcmp, which was used for detailed decomposition. It is primarily intended for use in nonlinear decomposition and convenient methods for dealing with path dependency [30], as well as overcoming the identification problem associated with the selection of a reference category when dummy variables are included as Oaxaca. Multivariate decomposition (mvdcmp) determines the high-outcome group automatically and uses the low-outcome group as a reference [31].

### Ethical considerations

Even if ethical procedures were the responsibility of the institutions that carried out the survey, authors of this research obtained written permission letter for utilization of the dataset from the Measure DHS International Program which authorized for the data-sets. The written approval letter was obtained from the Measure DHS International Program which authorized for the data-sets. Before data collection EDHS data collection materials approved for compliance of the requirements of 45 CFR 46, "Protection of Human Subjects" by Institutional Review Board (IBR) of each countries. Confidentiality was maintained anonymously and the data was used solely for the purpose of the current study.

## Results

### Urban rural delivery distribution in the study areas

Fig 1 illustrates the overall, as well as the urban and rural, women's place of delivery in Ethiopia, Kenya, Ugandan and Tanzania. In Ethiopia only 27.43% women gave at health facility. The difference in health facility delivery between urban and rural residences was high (58.44%) Ethiopia. In Uganda, institutional delivery was relatively highest from selected study countries almost three-fourth (74.64%), and with the lowest urban-rural health facility delivery gap (17.75). However, urban-rural institutional delivery difference was highly statistically significant (Chi-square, p-value <0.001).

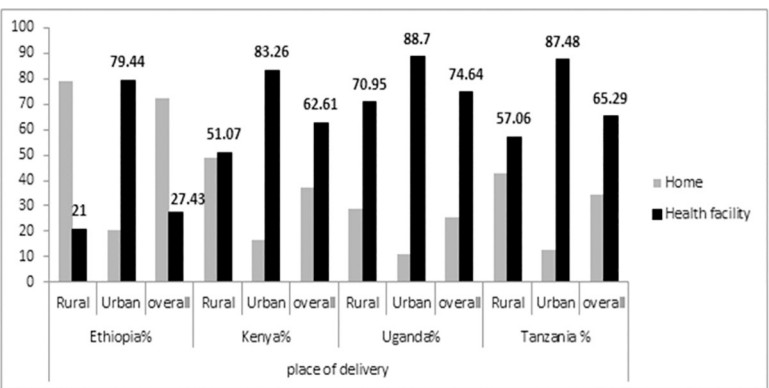

**Fig 1. Proportion of institutional delivery in Ethiopia, Kenya, Uganda and Tanzania stratified by their residency.**

Table 1 shows the differences in selected characteristics of households in the urban and rural regions in the four countries. The rural households, are on average, less educated, having lower access to media and at lower wealth status and distance of the health facilities is considered as a problem than urban households. While the urban-rural differentials are higher in the case of Ethiopia, for instance, 70.95% of women and 51.62% of husbands in rural resident have no formal education and 27.78% of women had no ANC follow up. The percentage of urban women with secondary or higher education is more than thirteen times of rural women in Ethiopia. However, this disparity was only two to three times in (Kenya, Uganda and Tanzania). Health facility distance was a big problem for 65.91% and 52.01% rural women in Ethiopia and Tanzania respectively. Ethiopia has the highest in not having ANC (41.15%). Regarding media access 87.44% rural Ethiopian and 61.56% Tanzanian households had no access (Table 1).

## Aggregate decomposition estimates

Table 2 presents the decomposition of the urban-rural institutional delivery gap into three components; a gap due to the difference in the level of characteristics, a gap due to the difference in the effect of the coefficients and a gap due to the interaction. The average total difference in predicted institutional delivery between the urban and rural groups was 0.58 for Ethiopia, 0.29 for Kenya, 0.18 for Uganda and 0.31 for Tanzania. The aggregate decomposition depicted that unexplained components were approximately 39.72% when using the Oaxaca decomposition, compared to 36.92% when using the Blinder decomposition, suggesting that discrimination to rural women contributes more to the gap in institutional delivery than favoritism of urban women in Ethiopia. These results are proven by the Cotton and Reimer decompositions, which are able to show how both discrimination and favoritism contribute to the gap. Certainly, using the Reimers decomposition, only 14.5% of the gap is explained by advantage to urban women, while 19.56% is explained by disadvantage to rural women. Likewise, when using the Cotton decomposition, solely 4.96% of the gap is explained by advantage to urban women, compared to 31.79% explained by disadvantage to rural women in Ethiopia.

Regarding Kenya aggregate decomposition portrayed that unexplained components were approximately 14% when using the Oaxaca decomposition, compared to 13% when using the Blinder decomposition, showing that discrimination to rural women contributes more to the gap in institutional delivery than favoritism of urban women in Kenya. This result is also parallel with the outcome obtained by the Reimer and Cotton decompositions. Certainly, using the Reimers decomposition, 5.48% of the gap is explained by advantage to urban women, while 7.03% is explained by disadvantage to rural women. Similarly, when using the Cotton decomposition, 3.62% of the gap is explained by advantage to urban women, compared to 9.11% explained by disadvantage to rural women in Kenya. The finding also gives almost similar result for Ugandan and Tanzania.

Across to the four countries the explained (endowment) component made the largest percentage of the gap, ranging from 59.24% (Uganda) to 87.34% (Kenya) for the Oaxaca decomposition and 47.35% (Uganda) to 86.42% (Kenya) for Blinder decomposition (Table 2).

## Detailed decomposition

**Difference in characteristics (covariate distribution).** Tables 3 and 4: demonstrates the contribution of individual characteristics to the institutional delivery gap in place of residence. These results showed a significant facility delivery gap in urban rural residencies (0.61, 0.32, 0.19 and 0.30, p < 0.001) in Ethiopia, Kenya, Uganda and Tanzania respectively. About 82%, 67%, 57% and 75% of the institutional delivery disparities were explained by the differences in

**Table 1. Baseline characteristics of reproductive-age women by place of delivery in selected East Africa countries, 2014–2016.**

| Characteristics | 2016 Ethiopia | | 2014 Kenya | | 2016 Uganda | | 2015 Tanzania | |
|---|---|---|---|---|---|---|---|---|
| residence | rural% | urban% | rural% | urban% | rural% | urban% | rural% | urban% |
| Proportion | 89.04 | 10.96 | 64.17 | 35.83 | 79.23 | 20.77 | 72.93 | 27.07 |
| mean age at birth | 27.4± 6.6* | 27.1± 5.5* | 26.7±6.6** | 25.9±5.7** | 26.47±6.8 | 26.15±6.2 | 27.21±7.25 | 26.73±6.4** |
| Education (%) | $chi^2(3),48^{**}$ | | $chi^2(3), 26^{**}$ | | $chi^2(3),32^{**}$ | | $chi^2(3),761^{**}$ | |
| No education | 70.81 | 26.20 | 15.37 | 5.39 | 12.25 | 6.29 | 25.23 | 9.28 |
| Primary | 26.24 | 32.29 | 62.32 | 44.86 | 67.35 | 40.02 | 66.36 | 60.74 |
| Secondary | 2.56 | 21.90 | 18.19 | 33.64 | 17.07 | 36.20 | 8.19 | 27.33 |
| Higher | 0.39 | 19.60 | 4.11 | 16.10 | 3.33 | 17.49 | 0.22 | 2.65 |
| Wealth index (%) | $chi^2(2),133^{**}$ | | $chi^2(2),98^{**}$ | | $chi^2(2),52^{**}$ | | $chi^2(2),62^{**}$ | |
| Poor | 51.61 | 7.18 | 61.21 | 13.75 | 51.75 | 14.84 | 59.28 | 8.24 |
| Middle | 23.01 | 2.04 | 22.04 | 10.92 | 22.33 | 7.96 | 24.34 | 5.37 |
| Rich | 25.18 | 90.78 | 16.75 | 85.33 | 25.92 | 77.12 | 16.13 | 86.39 |
| Marital status (%) | $chi^2(2),97^{**}$ | | $chi^2(2),22^{**}$ | | $chi^2(2),70^{**}$ | | $chi^2(2),111^{**}$ | |
| Not married | 0.42 | 1.34 | 6.81 | 7.51 | 3.93 | 6.21 | 4.21 | 9.22 |
| Married | 95.29 | 91.78 | 84.44 | 82.83 | 84.71 | 81.03 | 85.11 | 76.46 |
| Widowed | 4.29 | 6.88 | 8.74 | 9.66 | 11.36 | 12.77 | 10.69 | 14.31 |
| Distance from a health facility (%) | $chi^2(1),22^{**}$ | | $chi^2(1),405^{**}$ | | $chi^2(1),617^{**}$ | | $chi^2(1),238^{**}$ | |
| big problem | 65.91 | 17.53 | 35.29 | 14.25 | 46.10 | 21.30 | 52.01 | 33.54 |
| not a big problem | 34.09 | 82.47 | 64.71 | 85.75 | 53.90 | 78.70 | 47.99 | 66.46 |
| Pregnancy Characteristics | | | | | | | | |
| ANC visits | $chi^2(1),896^{**}$ | | $chi^2(1),134^{**}$ | | $chi^2(1),2.99$ | | $chi^2(1),0.51$ | |
| No ANC | 27.80 | 7.82 | 3.72 | 1.47 | 1.36 | 0.96 | 1.49 | 1.21 |
| one and above | 72.20 | 92.18 | 96.28 | 98.53 | 98.64 | 99.04 | 98.51 | 98.79 |
| Parity | $chi^2(2),582^{**}$ | | $chi^2(2),706^{**}$ | | $chi^2(2),302^{**}$ | | $chi^2(2),345^{**}$ | |
| 1 | 11.23 | 27.18 | 13.80 | 26.78 | 11.88 | 19.30 | 14.81 | 23.46 |
| 2–4 | 42.50 | 55.48 | 52.46 | 60.03 | 47.38 | 57.26 | 46.22 | 57.67 |
| 5 and above | 46.27 | 17.33 | 33.75 | 13.19 | 40.74 | 23.44 | 38.98 | 18.87 |
| Women occupation | $chi^2(3),20^{**}$ | | $chi^2(3),543^{**}$ | | $chi^2(3),26^{**}$ | | $chi^2(3),38^{**}$ | |
| not working | 57.60 | 40.56 | 29.81 | 32.91 | 15.37 | 23.68 | 11.74 | 26.76 |
| government employed | 10.26 | 37.89 | 6.15 | 13.22 | 12.27 | 32.32 | 1.30 | 6.91 |
| Agriculture | 27.15 | 10.54 | 34.38 | 9.08 | 53.79 | 16.16 | 72.43 | 15.51 |
| manual worker | 4.99 | 11.01 | 29.67 | 44.78 | 18.57 | 27.84 | 14.53 | 50.82 |
| Husband Education (%) | $chi^2(3),38^{**}$ | | $chi^2(3),654^{**}$ | | $chi^2(3),949^{**}$ | | $chi^2(3),754^{**}$ | |
| No education | 51.62 | 18.88 | 12.75 | 3.29 | 7.79 | 4.85 | 17.77 | 4.03 |
| Primary | 40.84 | 29.55 | 58.58 | 38.03 | 60.80 | 34.37 | 72.15 | 60.59 |
| Secondary | 5.74 | 24.17 | 22.23 | 38.64 | 23.88 | 35.43 | 9.27 | 28.29 |
| Higher | 1.80 | 27.40 | 6.44 | 20.04 | 7.54 | 25.35 | 0.81 | 7.09 |
| Husband occupation | $chi^2(3),46^{**}$ | | $chi^2(3),622^{**}$ | | $chi^2(3), 878^{**}$ | | $chi^2(3),28^{**}$ | |
| not working | 7.73 | 3.71 | 1.85 | 0.82 | 3.23 | 2.74 | 1.01 | 0.75 |
| government employed | 7.55 | 35.16 | 12.20 | 22.97 | 15.69 | 35.27 | 3.51 | 11.14 |
| Agriculture | 70.70 | 18.66 | 32.32 | 6.00 | 43.38 | 10.68 | 72.57 | 19.48 |
| manual worker | 14.03 | 42.47 | 53.63 | 70.21 | 37.71 | 51.31 | 22.91 | 68.63 |
| Birth order (mean) | 4.17 | 2.71 | 3.58 | 2.48 | 3.96 | 3.01 | 3.82 | 2.79 |
| media access | $chi^2(1),48^{**}$ | | $chi^2(1),889^{**}$ | | $chi^2(1),889^{**}$ | | $chi^2(1),561^{**}$ | |
| no access | 87.44 | 35.41 | 36.55 | 13.91 | 41.74 | 24.73 | 61.56 | 33.05 |

(*Continued*)

**Table 1.** (Continued)

| Characteristics | 2016 Ethiopia | | 2014 Kenya | | 2016 Uganda | | 2015 Tanzania | |
|---|---|---|---|---|---|---|---|---|
| residence | rural% | urban% | rural% | urban% | rural% | urban% | rural% | urban% |
| have accesses | 12.56 | 64.59 | 63.45 | 86.09 | 58.26 | 75.27 | 38.44 | 66.95 |

** Significant <0.01

* Significant at <0.05 and chi2 = chi-square.

distributions of characteristics (endowments) between urban and rural residences in Ethiopia, Kenya, Uganda and Tanzania respectively.

In Ethiopia the majority of the gap in institutional delivery was explained by wealth status difference between urban and rural women, poor wealth status (-36.55%) contributed for widening of the gap and rich wealth status (70.40%) contributed for narrowing of this gap. The distribution of antenatal care follow up (8.67%), women having one parity (5.89%), and having more than four parity (14.87%), having access to mass media (8.57%), husband's secondary (4.01%) and higher (10.26%) education level were factors that helping to achieve narrowing of

**Table 2. Three fold decomposition estimates of place of delivery by residence in the selected countries.**

| Country | Ethiopia | | Kenya | | Uganda | | Tanzania | |
|---|---|---|---|---|---|---|---|---|
| Decomposition | Coefficient | Percentage (%) | Coefficient | Percentage (%) | Coefficient | Percentage (%) | Coefficient | Percentage (%) |
| **Oaxaca Decomposition weight = 1** | | | | | | | | |
| Explained | 0.371 | 55.18 | 0.252 | 87.34 | 0.107 | 59.24 | 0.230 | 73.66 |
| Unexplained | 0.234 | 36.39 | 0.039 | 13.58 | 0.096 | 52.65 | 0.070 | 22.36 |
| Interaction | -0.016 | 8.44 | -0.003 | -0.93 | -0.022 | -11.88 | 0.012 | 3.98 |
| Total | 0.589 | 100 | 0.288 | 100 | 0.181 | 100 | 0.313 | 100 |
| **Blinder Decomposition: Weight = 0** | | | | | | | | |
| Explained | 0.355 | 63.62 | 0.249 | 86.42 | 0.086 | 47.35 | 0.243 | 77.65 |
| Unexplained | 0.217 | 44.82 | 0.037 | 12.66 | 0.074 | 40.76 | 0.082 | 26.34 |
| Interaction | 0.017 | -8.44 | -0.003 | 0.93 | 0.022 | 11.88 | -0.012 | -3.98 |
| Total | 0.589 | 100 | 0.288 | 100 | 0.181 | 100 | 0.313 | 100 |
| **Reimers Decomposition: Weight = 0.5** | | | | | | | | |
| productivity | 0.388 | 64.70 | 0.252 | 87.49 | 0.076 | 53.77 | 0.242 | 77.41 |
| Advantaged | 0.085 | 17.33 | 0.016 | 5.48 | 0.032 | 17.57 | 0.033 | 10.49 |
| Disadvantage | 0.115 | 17.97 | 0.020 | 7.03 | 0.052 | 28.67 | 0.038 | 12.09 |
| **Total** | 0.589 | 100 | 0.288 | 100 | 0.181 | 100 | 0.313 | 100 |
| **Cotton: De** | **weight = 0.79** | | **weight = 0.65** | | **weight = 0.82** | | **weight = 0.76** | |
| Productivity | 0.372 | 65.09 | 0.252 | 87.27 | 0.090 | 49.76 | 0.244 | 77.86 |
| Advantage | 0.029 | 6.13 | 0.010 | 3.62 | 0.011 | 5.90 | 0.014 | 4.49 |
| Disadvantage | 0.187 | 28.78 | 0.026 | 9.11 | 0.081 | 44.35 | 0.055 | 17.65 |
| Total | 0.589 | 100 | 0.288 | 100 | 0.181 | 100 | 0.313 | 100 |
| **Neumark (pooled regression coefficients)** | | | | | | | | |
| Productivity | 0.478 | 72.04 | 0.264 | 91.45 | 0.116 | 63.87 | 0.252 | 81.64 |
| Advantage | 0.123 | 24.41 | 0.015 | 5.25 | 0.051 | 28.27 | 0.042 | 13.30 |
| Disadvantage | 0.017 | 3.53 | 0.010 | 3.30 | 0.014 | 7.85 | 0.016 | 5.06 |
| Total | 0.589 | 100 | 0.288 | 100 | 0.181 | 100 | 0.313 | 100 |

De = decomposition.

**Table 3.  Detailed decomposition of institutional delivery by place of residence for Ethiopian and Kenyan women.**

| Country | Ethiopia | | | Kenya | | |
|---|---|---|---|---|---|---|
| Decomposition | estimate | se | Percent | estimate | se | Percent |
| Raw difference | 0.608** | 0.017 | 100 | 0.318** | 0.011 | 100 |
| Explained | 0.498** | 0.046 | 81.92 | 0. 214** | 0.015 | 67.2 |
| Unexplained | 0. 110* | 0.052 | 18.08 | 0.015** | 0.021 | 32.8 |
| **Endowment (Explained component) = difference in characteristics (E)** | | | | | | |
| wealth index | | | | | | |
| Poor | -0.222** | 0.042 | -36.55 | -0.002 | 0.012 | -0.61 |
| Middle | ref | | | ref | | |
| Rich | 0.428** | 0.0565 | 70.40 | 0.118** | 0.016 | 37.30 |
| Distance from HF | | | | | | |
| Big problem | 0.005 | 0.025 | 0.81 | -0.0001 | 0.005 | 0.04 |
| Not a big problem | reference | | | ref | | |
| Parity | | | | | | |
| 1 | 0.036** | 0.010 | 5.89 | 0.009* | 0.004 | 2.95 |
| 2–4 | ref | | | ref | | |
| 5 and above | 0.091** | 0.016 | 14.87 | 0.019** | 0.007 | 5.88 |
| Age at birth | 0.005** | 0.001 | 0.85 | -0.003 | 0.002 | -1.14 |
| ANC follow up | | | | | | |
| No | reference | | | ref | | |
| Yes | 0.053** | 0.015 | 8.67 | 0.004** | 0.001 | 1.31 |
| Husband occupation | | | | | | |
| Not working | ref | | | ref | | |
| Government employed | -0.028 | 0.017 | -4.58 | 0.01 | 0.01 | 3.02 |
| Agriculture | 0.051 | 0.032 | 8.37 | -0.008 | 0.014 | -2.54 |
| Manual worker | -0.026 | 0.015 | -4.33 | 0.013 | 0.031 | 4.09 |
| Husband education | | | | | | |
| no education | ref | | | ref | | |
| Primary | -0.006 | 0.005 | -0.96 | 0.014* | 0.007 | 4.39 |
| secondary | 0.024* | 0.012 | 4.01 | -0.000 | 0.006 | -0.03 |
| higher | 0.062** | 0.017 | 10.26 | 0.018* | 0.007 | 5.75 |
| Media exposure | | | | | | |
| No access | reference | | | | | |
| Have access | 0.052** | 0.023 | 8.57 | | | |
| Women education | | | | | | |
| No education | ref | | | ref | | |
| Primary | 0.004 | 0.003 | 0.66 | -0.008 | 0.005 | -2.55 |
| Secondary | 0.030 | 0. 015 | 4.99 | 0.014* | 0.006 | 4.52 |
| Higher | 0.030 | 0.021 | 4.87 | 0.035** | 0.008 | 10.88 |
| Mothers occupation | | | | | | |
| Not working | | | | ref | | |
| Government employed | | | | -0.003 | 0.003 | -0.83 |
| Agriculture | | | | -0.005 | 0.008 | -1.73 |
| Manual worker | | | | -0.001 | 0.004 | -0.44 |
| **unexplained (Due to difference in coefficients (C))** | | | | | | |
| Wealth | | | | | | |
| Poor | 1.02 | 1.31 | 167.78 | 0.047* | 0.021 | 14.88 |
| Middle | ref | | | ref | | |

(*Continued*)

**Table 3.** (Continued)

| Country | Ethiopia | | | Kenya | | |
|---|---|---|---|---|---|---|
| Decomposition | estimate | se | Percent | estimate | se | Percent |
| Rich | 0.571 | 0.741 | 94.01 | 0.027** | 0.006 | 8.52 |
| Distance from HF | | | | | | |
| Big problem | 0.125 | 0.155 | 20.60 | -0.031 | 0.019 | -9.87 |
| Not a big problem | ref | | | ref | | |
| Parity | | | | | | |
| 1 | 0.0145 | 0.015 | 2.39 | -0.009 | 0.005 | -2.75 |
| 2–4 | Ref | | | ref | | |
| 5 and above | -0.308 | 0.408 | -50.68 | 0.017 | 0.014 | 5.21 |
| Age at birth | 1.21 | 1.719 | 199.35 | | | |
| ANC follow up | | | | | | |
| No | Ref | | | ref | | |
| Yes | 0.108 | 0.255 | 17.76 | -0.015 | 0.09 | -4.76 |
| Husband occupation | | | | | | |
| Not working | Ref | | | ref | | |
| Government employed | -0.035 | 0.047 | -5.86 | 0.008 | 0.012 | 2.52 |
| Agriculture | -0.259 | 0.419 | -42.66 | 0.013 | 0.031 | 4.09 |
| Husband education | | | | | | |
| No education | ref | | | ref | | |
| Primary | 0.043 | 0.089 | 7.13 | -0.145** | 0.028 | -45.69 |
| Secondary | -0.001 | 0.013 | -0.22 | -0.043** | 0.012 | -13.50 |
| Higher | 0.005 | 0.008 | 0.87 | -0.010 | 0.010 | -0.029 |
| Media exposure | | | | | | |
| No access | Ref | | | | | |
| Have access | 0.032 | 0.047 | 5.41 | | | |
| Women Education | | | | | | |
| No education | Ref | | | ref | | |
| Primary | -0.012 | 0.039 | -2.01 | -0.017 | 0.026 | -5.40 |
| Secondary | 0.001 | 0.007 | 0.14 | -0.014 | 0.009 | -4.26 |
| Higher | 0.0003 | 0.002 | 0.040 | 0.004 | 0.004 | 1.11 |
| Mothers occupation | | | | | | |
| Not working | | | | ref | | |
| Government employed | | | | -0.004 | 0.004 | -1.15 |
| Agriculture | | | | 0.0004 | 0.014 | 0.12 |
| Manual worker | | | | -0.009 | 0.010 | -2.77 |

*Significant at p-value <0.05 confidence level

** Significant at p-value<0.01 confidence level, se = standard error HF = health facility, ref = reference category.

the urban-rural institutional delivery gap if these covariate distribution equalized to the level of urban women for rural women as well.

Rich wealth, both one and grand parity (>4), ANC follow up, and secondary and higher women's education, both primary and higher husband education in Kenya were factors act to narrow the institutional delivery gap between urban and rural residences (Table 2).

The rich wealth, four and above number of ANC visit, grand parity Uganda and rich wealth, higher husband education, and more than four parity in Tanzania were covariate distribution contributing for narrowing the urban rural institutional delivery differences (Table 4).

**Table 4. Detailed decomposition of institutional delivery by place of residence for Ugandan and Tanzanian women.**

| Country | Uganda | | | Tanzania | | |
|---|---|---|---|---|---|---|
| Decomposition | estimate | se | Percent | estimate | se | Percent |
| Raw difference | 0.192** | 0.010 | 100 | 0.303** | 0.13 | 100 |
| Explained | 0. 110** | 0.014 | 57.31 | 0.228** | 0.038 | 75.16 |
| Unexplained | 0.082** | 0.019 | 42.69 | 0.075 | 0.044 | 24.84 |
| **Endowment (Explained component) = difference in characteristics (E)** | | | | | | |
| Wealth | | | | | | |
| Poor | -0.003 | 0.013 | -1.47 | 0.049 | 0.032 | 16.20 |
| Middle | ref | | | ref | | |
| Rich | 0.052** | 0.020 | 27.19 | 0.097** | 0.036 | 32.23 |
| Age at birth | -0.0003 | 0.0002 | -0.21 | -0.004 | 0.002 | -1.39 |
| Number of ANC | | | | | | |
| ANC< = 3 | ref | | | ref | | |
| ANC> = 4 | 0.006** | 0.002 | 2.91 | 0.005 | 0.006 | 1.49 |
| Women Education | | | | | | |
| No education | ref | | | ref | | |
| Primary | 0.003 | 0.012 | 1.49 | 0.025 | 0.04 | 8.33 |
| Secondary | 0.010 | 0.010 | 5.59 | -0.004 | 0.007 | -1.19 |
| Higher | 0.005 | 0.012 | 2.68 | - | - | - |
| Husband education | | | | | | |
| No education | ref | | | ref | | |
| Primary | 0.010 | 0.013 | 5.32 | 0.005 | 0.008 | 1.47 |
| Secondary | -0.002 | 0.006 | -1.02 | 0.007 | 0.013 | 2.17 |
| Higher | 0.009 | 0.012 | 4.64 | 0.015* | 0.007 | 4.83 |
| Parity | | | | | | |
| 1 | 0.006 | 0.003 | 3.15 | 0.003 | 0.003 | 0.84 |
| 2–4 | ref | | | ref | | |
| 5 and above | 0.013* | 0.005 | 6.89 | 0.036** | 0.009 | 11.92 |
| **unexplained (Due to difference in coefficients (C))** | | | | | | |
| Wealth | | | | | | |
| Poor | 0.019 | 0.027 | 10.14 | 0.030 | 0.043 | -9.89 |
| Middle | ref | | | ref | | |
| Rich | 0.022 | 0.014 | 11.62 | 0.009 | 0.012 | 3.05 |
| Age at birth | 0.211* | 0.099 | -17.60 | | | |
| Number of ANC | | | | | | |
| ANC< = 3 | ref | | | ref | | |
| ANC> = 4 | -0.016 | 0.022 | -8.40 | -0.021 | 0.020 | -7.22 |
| Women Education | | | | | | |
| No education | ref | | | ref | | |
| Primary | -0.007 | 0.045 | -3.88 | 0.025 | 0.04 | 8.33 |
| Secondary | 0.001 | 0.014 | 0.71 | -0.004 | 0.007 | -1.19 |
| Higher | -0.009 | 0.005 | -4.84 | - | - | - |
| Husband education | | | | | | |
| No education | ref | | | ref | | |
| Primary | -0.037 | 0.046 | -19.37 | -0.083 | 0.059 | -27.34 |
| Secondary | -0.022 | 0.021 | -11.43 | -0.0031 | 0.010 | -1.03 |
| Higher | -0.003 | 0.009 | -1.34 | 0.004* | 0.002 | 1.37 |
| Parity | | | | | | |

*(Continued)*

**Table 4.** (Continued)

| Country | Uganda | | | Tanzania | | |
|---|---|---|---|---|---|---|
| Decomposition | estimate | se | Percent | estimate | se | Percent |
| 1 | 0.006 | 0.009 | 2.88 | -0.011 | 0.012 | -3.65 |
| 2–4 | ref | | | ref | | |
| 5 and above | -0.033 | 0.019 | -17.60 | -0.047 | 0.025 | -15.56 |

*Significant at 90% confidence level

** Significant at 95% confidence level, se = standard error.

**Difference due to coefficients (the effect of covariate differences).** We found that differences in effects (difference due to in coefficients) account for 18.08%, 32.8%, 42.69% and 24.84% of the observed urban-rural differential in the place of delivery in Ethiopia, Kenya, Uganda and Tanzania respectively.

For Kenya, both the effect of poor (14.88%) and rich (8.52%) wealth status contribute for narrowing the gap of institutional delivery between urban and rural women if the rural women wealth increased from poor to middle level wealth and from middle to rich wealth status as their urban counterparts. Whereas, the effect of primary and secondary husband education status were responsible for widening of urban rural institutional delivery gap by 45.69% and 13.5% in place of residences if the rural women education improved from no formal education to primary and secondary level respectively.

For Uganda the effect of each one year increase in maternal age at child birth in rural residents increase the urban rural institutional delivery by 17.60%. For Tanzania the effect of higher husband education contributes 1.37% reduction urban rural institutional delivery disparities if rural Tanzanian husbands able to attend higher education as their rural counterparts (Table 4).

## Discussion

In low and middle-income countries, there is documented evidence that, averagely, urban women have higher health facility births compared to rural residents [4, 5, 9]. In fact, giving birth in a health facility can prevent maternal and infant deaths by providing timely skilled birth attendants and appropriate medications to address birth complications. In this study, we comprehensively investigated, and identified the underlying factors, behind the urban-rural gap in institutional delivery in Ethiopia, Kenya, Ugandan and Tanzania, using a Blinder-Oaxaca and related decomposition analysis, on which limited research has been conducted. To our knowledge, this is the first study to explain the observed gap in institutional delivery between urban and rural women in East African countries. This helps to inform and guide policies aimed at reducing women's health inequalities and improving population health in the study area.

We discovered significant urban-rural disparities across the study countries, though they were not uniform: the highest urban-rural institutional delivery disparity was documented in Ethiopia (58.44 percentage points (urban 79% and rural 21%), 32.19 in Kenya, 30.42% in Tanzania, and the lowest disparity was observed in Uganda (17.75%) (Fig 1). This variation could be due to rural women's poor wealth status, low antenatal care coverage, husband illiteracy and less access to media in Ethiopia, as confirmed by our descriptive statistics that almost a third (28%) of pregnant women have no any ANC follow-up, and more than 87% of women have no access to mass media as compared to 92.18–99% women had at least one ANC follow

up in urban settings and 65–86% had access to media at least one per week. It is obvious that rich women in urban areas have easy access to information regarding maternal and child health through different media, which helps them to have ANC follow up and finally to give birth at a health facility [32].

Our findings show that, when the coefficient effect is held constant, more than half of the observed disparities in institutional delivery among urban and rural women in the study area can be attributed to differences in the distribution of institutional delivery covariates between these groups, regardless of the decomposition type. Meaning that, contribution of composition (endowments) changes were more important than behaviour (coefficient) changes to reduce the gap between urban-rural women's institutional delivery. This could be because urban women are more likely to give birth in a health facility than rural counterparts because they are better endowed with the factors that encourage institutional delivery (rich wealth, ANC, access to media, both maternal and husband education). This explanation was supported by our descriptive result (Table 1), which revealed that 90.78% of urban women compared to 25.8% of rural women were rich in wealth, and rich wealth status with its positive coefficient (Tables 3 and 4) implying that if rural women wealth reached the level of urban women counterparts, the gap in institutional delivery would be narrowed. This was also true for ANC follow up, presence of media access, single and grand parity. This finding is also supported by results in India [10] in Ghana [11] and in 80 low and middle-income countries study result [33]. In addition, husband secondary and higher education and age at childbirth were all important contributors to close the urban-rural gap of institutional delivery in Ethiopia. Secondary and higher education for women, as well as primary and secondary education for husbands, were also important factors in Kenya. Factors such as Ethiopia's low wealth status, Kenya's primary and secondary husband education, and Uganda's maternal age at childbirth all contributed to the widening of the gap. These findings are consistent with previous studies that identified wealth [10, 12, 17, 20], antenatal visits [22], and parity [16, 22] as important predictors of institutional delivery, and are supported by our descriptive analysis, which depicted women in wealthy households, a higher proportion of women attending antenatal care visits, and lower parity among urban women compared to rural women (Table 1).

About 19.98% and 2.25% proportional covariate change in antenatal care follow up between urban and rural women in Ethiopia and Kenya (Table 1) contribute to 8.67%, and 1.31% change in institutional delivery (Table 3) between urban and rural women respectively. The possible reason might be the fact that women who had at least first antenatal follow-up were more informed about the benefits of institutional delivery [34] and more likely to deliver at health facilities than those who had no follow-up at all [22, 34]. And the service (ANC) is predominantly distributed in urban settings with public and private options.

In Ethiopia, 65.60%, Kenya 68.58%, Uganda 51.20% and Tanzania 70.26% urban-rural proportional change in rich wealth contributed to the change of 70.40%, 37.30%, 27.19% and 32.23% change in institutional delivery between urban-rural settings respectively. Despite the fact that the governments of each county provided free maternal and child services in order to expand the opportunities for institutional delivery [35, 36], a significant number of rural women chose home delivery due to financial constraints. This could be because women from low-income households may not have access to antenatal care follow-up due to daily agronomic duties in rural areas, as well as dispersed health facilities to easily access maternity services. Poor rural women may be more likely to give birth at home due to a lack of a substituted role for women and the difficulty of accessing health care services [37, 38]. Moreover, women in poor rural households are uneducated to be informed on the benefit of institutional delivery as supported by our descriptive results (Table 1) and other studies [38–40]. This implies that improving the economic status of rural women by itself could improve institutional delivery

from 27.19% to more than 70% and the population's health at large. Moreover, if poor rural women's wealth equalized to urban middle wealth level, institutional delivery will be improved more than urban middle-wealth status women by about 36.55%. This implies wealth status improvement interventions in rural settings, particularly in poor households, significantly contribute to the improvement of institutional delivery. On the other hand, rural middle-class mothers are treated the same as urban rich women, and institutional delivery in rural areas will be increased to match but not exceed the level of urban women's institutional delivery. This is in agreement with our argument that poor wealth in rural women is a major problem for getting time for ANC and other important information to make decisions on place of delivery than middle and rich wealthy women in urban settings. Whereas, in middle wealth status, women in urban areas, institutional delivery is less affected by wealth difference [29].

For Uganda, on average, the one year age difference at the birth of a child between urban and rural women contributed to a 17.60% change in institutional delivery. If rural women had a similar age at child birth as urban women, the 17.6% rural-urban institutional delivery gap would be increased. This implies the age of the mother for urban women at birth is less sensitive than rural women to give birth at a health facility.

Keeping endowment characteristics in mind, about 18.08%, 32.8%, 42.69%, and 24.84% of the difference in urban-rural institutional delivery in Ethiopia, Kenya, Uganda, and Tanzania was contributed by behavior change (due to differences in coefficients) to the place of delivery. Significant positive contributions to behavior change in terms of rich wealth for Kenya, and higher husband education in Tanzania were noted. As a result, not only does the covariate effect of rich wealth influence the institutional delivery gap, but the behavioral effect of rich wealth also plays a significant role in narrowing the urban rural gap by raising the wealth of rural women to that of their urban counterparts, and this finding is supported by findings in Ghana [11].

The positive coefficients of the unexplained component suggest that a change in covariates that increases institutional delivery would lead to insignificant improvement in rural areas' institutional delivery compared to urban areas since behavior change enhancing intervention are predominantly important. So, for Kenyan women, for example, approximately 32.8% of the urban-rural institutional delivery disparity will be reduced not only by improving rural women's income (wealth status), but also by increasing women's understanding of the benefits of institutional delivery through broad coverage of women's education, prenatal care, and access to information on maternal health issues [11, 17, 23]. Unexplained component contribution was particularly high in Uganda and Kenya (42.69% and 32.8%) respectively, implying that governments and policymakers should place a greater emphasis on behavioral and awareness creation interventions in rural women to close the gap between the two groups, in addition to accessing determinant factors that increase institutional delivery.

In the current study, results of the Blinder-Oaxaca decomposition suggest that the rural-urban gap in institutional delivery in all countries is primarily explained by the difference in the magnitude of the determinants of institutional delivery (covariate effect is dominant), rather than by differences in their effect. In particular, wealth disparity between urban and rural households accounts for most of the overall gap. This is consistent with numerous study findings in Ghana [11], Ethiopia [23], Malawi which demonstrated socio-economic status associations with institution delivery. This implies that redistribution of wealth, and improving antenatal care and increasing media access for women in rural areas would be effective to reduce the residential inequalities in institutional delivery and improve maternal and child health in rural areas. However, in Uganda and Kenya, while the difference in determinants is significant and accounts for a large portion of the health facility delivery gap, the difference in the effect of covariates (coefficient effects) also plays a significant role in determining the

institutional delivery gap between urban and rural women. This is also concurrent with findings in Ghana [11] and Kenya [17]. According to these studies, education is a major determinant of maternal health care utilization, and education improves the ability to weigh the costs and benefits of health care facility services, as well as properly understand and conform to health messages. For these reasons, the redistribution of wealth, media access, controlling parity and overall upgrading in the level of the determinants in rural areas may not be adequate to achieve urban-rural institutional delivery. Accordingly, behavioral and awareness packages that would improve the effect of the determinants would be required interventions to narrow the rural-urban gap in institutional delivery inequalities.

## Limitations, strength and generalizability

The current study is not free from limitations. One limitation is the cross-sectional nature of the study design limits the ability to establish temporal relationship between time-varying variables. Second, both the dependent and independent variables were self-reported and are likely to have reporting and recall bias. However, since the survey data were focused on births, which are too big events, to be forgotten soon, we expect the accuracy of the data to be acceptable.

Third, the data analyzed here is relatively outdated; thus, the findings may not exactly reflect the current situation in study countries with regards to institutional delivery. Finally, important variables that will change the ratio of unexplained portions like region due to their inconsistent classification in the study area and other variables (like husband and mother-in-low decision in the place of delivery) due to their absence were omitted from analysis, leading to variable omitting bias. Despite these limitations, this study is important in that it gives an understanding and quantification of the drivers and magnitude of institutional delivery inequalities in urban-rural settings. The finding also generalized to similar countries as the data were representative and quality assured.

## Conclusions

The urban-rural institutional delivery disparities were high in study countries, particularly in Ethiopia. Covariate effects of institutional delivery explained a larger portion of the difference in institutional delivery rates between rural and urban women in these countries. Wealth, antenatal care, husband education, and access to the media were all significant factors. Future intervention to encourage institutional deliveries in rural women of these countries should therefore emphasis on increasing rural women income, access health care facilities to facilitate antenatal care utilization and to limit their children. However, unexplained components (effect of determinants) were significantly higher in Uganda, Kenya and Ethiopia, calling for integrated interventions on behavioral and awareness packages that would improve the effect of the determinants would be necessary interventions to close the rural-urban gap in institutional births. Researchers are also recommended to explore effective strategies to improve rural women's prosperity.

## Acknowledgments

We would like to extend our deepest gratitude to measure DHS for their permission to use the dataset.

## Author Contributions

**Conceptualization:** Reta Dewau, Dessie Abebaw Angaw, Baye Dagnew, Enyew Dagnew Yehuala.

**Data curation:** Reta Dewau, Baye Dagnew, Yigizie Yeshaw, Amare Muche, Dejen Getaneh Feleke, Eshetie Molla, Enyew Dagnew Yehuala.

**Formal analysis:** Reta Dewau, Dessie Abebaw Angaw, Getahun Molla Kassa, Melaku Yalew, Anissa Mohammed Hassen, Tadesse Awoke Ayele.

**Methodology:** Reta Dewau, Getahun Molla Kassa, Yigizie Yeshaw, Dejen Getaneh Feleke, Eshetie Molla, Melaku Yalew, Assefa Andargie, Wolde Melese Ayele, Bezawit Adane, Metadel Adane, Tadesse Awoke Ayele.

**Project administration:** Sisay Eshete Tadesse.

**Software:** Reta Dewau, Getahun Molla Kassa, Yigizie Yeshaw, Amare Muche, Melaku Yalew.

**Supervision:** Dessie Abebaw Angaw, Ahmed Hussien Asfaw, Yitayish Damtie, Mogesie Necho.

**Visualization:** Baye Dagnew, Amare Muche, Enyew Dagnew Yehuala, Sisay Eshete Tadesse, Ahmed Hussien Asfaw, Assefa Andargie, Muluken Genetu Chanie, Wolde Melese Ayele.

**Writing – original draft:** Reta Dewau.

**Writing – review & editing:** Dessie Abebaw Angaw, Getahun Molla Kassa, Baye Dagnew, Yigizie Yeshaw, Dejen Getaneh Feleke, Eshetie Molla, Sisay Eshete Tadesse, Melaku Yalew, Zinabu Fentaw, Ahmed Hussien Asfaw, Assefa Andargie, Muluken Genetu Chanie, Wolde Melese Ayele, Anissa Mohammed Hassen, Yitayish Damtie, Foziya Mohammed Hussein, Zinet Abegaz Asfaw, Elsabeth Addisu, Bezawit Adane, Fanos Yeshanew Ayele, Bereket Kefale, Aregash Abebayehu Zerga, Tefera Chane Mekonnen, Mogesie Necho, Oumer Abdulkadir Ebrahim, Metadel Adane, Tadesse Awoke Ayele.

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
