## [Decision Letter · Decision Letter 0]

26 Jan 2021

PONE-D-20-30414

Urban-rural disparities in institutional delivery among women in East Africa: a decomposition analysis

PLOS ONE

Dear Dr. Dewau,

Thank you for submitting your manuscript to PLOS ONE. After careful consideration, we feel that it has merit but does not fully meet PLOS ONE’s publication criteria as it currently stands. Therefore, we invite you to submit a revised version of the manuscript that addresses the points raised during the review process.

The reviewers have raised a number of concerns with the manuscript, including the length of the manuscript, the English language usage, insufficiently detailed methodology and the lack of sufficient statistical testing. The reviewers' comments can be viewed in full, below.

We look forward to receiving your revised manuscript.

Kind regards,

Natasha McDonald, PhD

Associat Editor

PLOS ONE

"No."

6. Please include a copy of Table 5 which you refer to in your text in lines 380 and 707.

7. Please include your tables as part of your main manuscript and remove the individual files. Please note that supplementary tables (should remain/ be uploaded) as separate "supporting information" files

8. Thank you for submitting the above manuscript to PLOS ONE. During our internal evaluation of the manuscript, we found significant text overlap between your submission and the following previously published works.

- https://equityhealthj.biomedcentral.com/articles/10.1186/s12939-019-1025-z

- https://healtheconomicsreview.biomedcentral.com/articles/10.1186/s13561-016-0097-3

- http://oar.icrisat.org/11163/1/Murendo%20and%20Murenje%202018.pdf

- https://bmjopen.bmj.com/content/10/9/e034786

Please revise the manuscript to rephrase the duplicated text, cite your sources, and provide details as to how the current manuscript advances on previous work. Please note that further consideration is dependent on the submission of a manuscript that addresses these concerns about the overlap in text with published work.

We will carefully review your manuscript upon resubmission, so please ensure that your revision is thorough

Reviewers' comments:

Reviewer's Responses to Questions

**Comments to the Author**

1. Is the manuscript technically sound, and do the data support the conclusions?

Reviewer #1: Yes

Reviewer #2: Yes

2. Has the statistical analysis been performed appropriately and rigorously? 

Reviewer #1: Yes

Reviewer #2: No

3. Have the authors made all data underlying the findings in their manuscript fully available?

Reviewer #1: Yes

Reviewer #2: Yes

4. Is the manuscript presented in an intelligible fashion and written in standard English?

Reviewer #1: Yes

Reviewer #2: No

5. Review Comments to the Author

Reviewer #1: Factors affecting the institutional delivery in LMIC is an important topic to explore. The authors used an interesting method to quantify the phenomenon. As authors stated the paper will be the first to use the Oaxaca-Blinder decomposition to examine the urban rural disparity in institutional delivery in selected east African countries

The paper is too long and most of information is not needed here or can be shortened

Background:

The last paragraph of the background “page 12 lines 112-121”) can be moved to methods section

Methods:

- Study population is just the women that have delivered in the past 5 years.

- Question: if a woman had delivered more than once in the past 5 years, please explain how are you exploring the information: are you considering all deliveries or just the last one?

Page 14, line: 148 delete “cite”

Results section:

Table 1 needs some statistical tests to these differences..

Table 2 needs some statistical tests and also there is a big problem for percentage of the “pregnancy-related deaths” (0.8%). The PRD is not affecting old women and most of deceases are women 60 years and above. You should recalculate this proportion based only on women of reproductive age (15-49years). This proportion will be much high that what the authors published. I think it will be more than 3%.

Please revise table 1 and table 2. The description in the manuscript is too confusing. I think on page 12, line 261 it is probably Table 2 and line 265 it is Table 1. Please review it again.

Table 1: Why not can provide the percentages instead of numbers? Too confusing.

What is the difference between table 1 and figure 1. I think they are providing the same information. If yes select only one of them.

Minors

The paper needs a deeper editing to correct several words: for instance:

Page 9, line 48: correct “differece” to “difference”

Page 10, line 65: correct “bellow” to “below”

Page 10, line 69: delete “(cite)”

The titles of the tables and figures should be more explicit.

Overall: the paper is too long and I will recommend the authors to shorten it by deleting or reducing several sections. For instance the last paragraph of the background is the same information that was described in the method. The description of the decomposition method is well known and can be shortened and authors can just provide references as needed. The results section can describe briefly the most important results and not provide too much information. At the end the readers are lost. Also the results sections should be shortened by describing results instead of providing some explanations in the results section with references in this section. All explanations and reference in the results section should be moved to Discussion section

Reviewer #2: - The authors have attempted to study drivers of urban-rural gap in facility delivery in four countries in East Africa. The line of research is interesting and has the potential to add something to knowledge on this area. However, the paper requires major and substantial revision to get most out of it. Substantial improvement in language is required to improve the readability of the paper. Every section of the paper needs revision. I put some suggestions (major and minor) according to the different sections of the manuscript.

Abstract

method

-indicate that the study was in four east African countries

result

-“covariate effect was dominate”; this contradicts with the result in the body

Conclusion

-“regional inequality” is confusing;

-“facilitate antenatal care utilization” is not clear, which ANC care? Four or more? Be specific

-“plan number of children” is vague

Introduction/background

-this section is an awkward and poorly written. You incoherently moved between different ideas; you begin with mortality, and then jumped to facility delivery and then back to mortality. Pls rewrite this section and ensure that the transitions between paragraphs are effective and clear. Also, you need to further substantiate it by adding points directly related with urban-rural disparity in facility delivery. Do not forget that you still follow certain chronological flows.

- in the “maternal mortality rate below 70”, change rate into ratio

-Lines 112-120: take this to method section

Methods

-DHS involves women 15-49 and men 15 to 59 as well as children under five. You need to mention this in the method section through your analysis in this study has been restricted to only women. You need to also clarify the distinction between “study population” and “sample”. All women in the reproductive age group in the selected households are samples, not study populations, unlike what you have said. Pls avoid the confusion around these terms

- the “sampling and sample size” section is too vague to understand how and how many enumeration areas (ESs) and participants were enrolled into the study. It needs to be clearly expounded that the DHS follows a stratified two-stage cluster design, where, following stratification by both urban and regions/districts (you just mentioned that stratification was based solely on urbanization and that is false), eligible women were selected in two stages. In this section, you can explain that succinctly. How many EAs were sampled in the first stage? How many households were sampled in each EA? What was the household and individual response rates? What is primary sampling unit/cluster and SSU? Explain

-what are the secondary dependent variables, since you have mentioned that facility delivery was the primary outcome variable? In this section, there some author notes that you should have avoided (non-cited statements)

- place of residence is the stratifier variable, not an exposure/independent variable. It is a variable the inequality was measured by accounting for the influences of exposure variables

-maternal age (age during the time of the surveys administrations) could not be an exposure variable, because it comes long after the outcome variable occurs. The word “cautious” is ambiguous in the “age at birth”

-it is strange that you did not explain how these variables were selected, and no citation to prior similar works was made. You need to tell us that these variables show certain theoretical and or statistical relationship with the intended response variables.

-region is missing from being an exposure variable

- distance from health facility: if this variable was created based on the variable v467d, then that is incorrect. This variable reflects access problem at the time of the survey for a woman's medical help, and is not related with facility delivery.

-how was media exposure variable was created and produced? Explain that to allow replication for other studies as well as to improve transparency. This applies also to other variables.

-marital status variable was measured in DHS at the time of the survyes, and might not reflect marital status of the women before becoming pregnant, and there fore might not be an exposure variable for facility delivery; it has to come before or at least as the same time as the outcome variable to call it an exposure variable. Pls see this variable in the DHS guide for scrutiny.

Analysis

-how did you assess the normality and skewness of the variables? Explain that.

-you have described that the analyses were all ”population-weighted” and that “survey design” was taken into cognizant during analysis. This expression is very misleading for many reasons. First, the phrase “population-weighted” adds noting to understanding except that it brings confusion with it. You simply say you analysis is “weighted”. Second, you did not explain how your analysis should be weighted. Third, when we say, survey design is taken into account, we are saying that we are accounting for the disproportionate sampling (that we weight our analysis) and the stratification and clustering effect are taken into account to produce nationally representative estimates. So, the later expression, that is, the survey design is taken into account is enough and there is no any reason to repeat that analyses were weighted as far as you have mentioned that you have already mentioned that the complex nature of the data has been accounted for. Finally, when you say survey design was accounted for, I am not sure whether you have accounted for both stratification and clustering, and if you have missed on of them, then the estimated Standard errors were most likely be erroneous.

-it is not clear why you have also used the decomposition techniques suggested by powers et al after you have calculated BO decomposition? Both methods can be used of binary variables. But the problem with BO is that results could hugely differ depending on the reference category chosen during analysis, aka “identification problem”. So, with BO, YOU must do normalization to make the finding consistent with the chosen reference categories of categorical variables.

- in the ethics section, please also mention that ethical procedures were the responsibility of the institutions that carried out the survey.

Results

urban rural delivery distribution in the study areas

-Since your outcome is facility delivery, just report that; reporting home deliveries adds only confusion

- the “delivery rate” is inappropriate usage as rate has a different meaning and does not go with delivery

-the statement “however…, Table-1” is very vague and I cannot understand it

- the statement “Kenya has the highest 249 (35.83%) urban resident women” is not clear

-lines 251-261: the texts here do not correspond to the content of table 1. Presented in table 1 is completely different from what the texts talks about. The texts about the differentials of urban and rural need to be presented well in terms of all the exposure variables so that we can see where these variables dominate, and this would then help us interpret the decomposition findings presented table 2 through table 4.

-Lines 262-265: it is the replication of table 2; pls avoid repetitions and redundancies

-Lines 287-288: I think the contributions of the variables in the decomposition can be to either widen or narrow down the gap; this is the most interesting portion of the findings and hence requires further explanation in the discussion section. Why some variables contribute in terms of widening the gap, but not others?

-lines 290-292: take this into the discussion

-Lines 249-356: this paragraph actually lacks clear interpretations/presentations of findings. When you say that disparity could be reduced if urban and rural settings have similar levels of wealth, for instance, it can also be interpreted that disparity can be lowered if wealth in urban areas is reduced to a level in rural settings, which is not actually the case. So, you need to clearly present that when wealth in rural settings are increased to a level in urban areas, urban-rural disparities can be narrowed by certain amount. Also, make sure that in this text, you have not repeated what you have already presented in the Table.

-Lines 357-359: in this small paragraph, you attempted to attach meaning to the observed contributions of the exposure variables. You are interpreting that variables with positive contribution like wealth tend to narrow down the disparity. However, how do these variables contribute this way? How do I know that these variables narrow, not widen the gap? I rather interpret the opposite way. I strongly recommend you to first document where these variables dominate (in urban or rural areas). For instance, if the richest and richer categories of wealth are predominantly found in the urban area, and if these categories have the capacity to increase the chance of giving birth in facilities, then surely, wealth tends to widen the disparity, rather than narrowing. Also, it is extremely important to show the contributions of each of the five categories of wealth, and of other categorical variables, and without doing that, you cannot interpret whether a variable contributes by widening or narrowing. At currently stands, wealth has 16% contribution to the disparity in Ethiopia, and we cannot know whether this contribution is to close the gap or not. My same comment applies to the other similar findings in the subsequent lines

Discussion

-Lines 408-410: revise the vague statement

-Lines 414-415: Even though there might not be a study on these countries together, there could be a number of studies that investigated facility delivery inequality in these countries, so unless you are 100% certain that there are no previous study, you should rather avoid making such bold statement that there were no prior studies.

-Lines 420-427: you need to show that the variation has been explained by the factors whose contributions have been explored and presented in the tables in the result section. For instance, in these lines, you have mentioned that the urban-rural disparity has been partly explained by the disparity in formal education and at least one ANC (or the lack thereof). However, in the result sections, you did not show separately that such variables have contributed to the disparity. Also, the word “ignorance” has nothing to do with your finding and seems also unethical.

-lines 433-435: it is important how the unequal distributions of endowments encourage urban women to attend to facility delivery. Which of these variables encourage urban women to deliver at facility, which variables discourage them? Explain the mechanism of actions of these variables to benefit women in urban and rural settings.

-lines 437-440: avoid the repetitions. How these variables widen the urban-rural gap?

-lines 446-448: not clear, and needs some clarification to help the statement read well

-lines 448-450: this interpretation is unclear; the evidence in the result section does not support the interpretations in these lines; how do we know that more frequent ANC visits could help urban women attend facility delivery in the absence of findings on the contributions of the specific categories ( that is 1-3 and >4= ANC visits) of the number of ANC visit?

-for the subsequent paragraphs, there is a need to make a major revision and make sure that every piece of your interpretation is completely informed by your findings. Since the analysis is incomplete, this has seriously affected the interpretations of the findings in this section of the paper.

-Also, it is good if you briefly elaborate on the policy implications of the study.

-it is not correct to blame the study being cross sectional as limitations of the study; causality cannot be made by a single study even with RCT, and more importantly, temporality is not the problem in the study at least for some of the variables like education and wealth. One important limitation in this is that in the BO decomposition, findings differ depending on the chosen reference categories of categorical variable, and hence requires further adjustment using “normalization” process. But, you need to overcome this problem in the study to get valid results.

-I cannot assume that these findings are generalized to the east African community since these four countries do not represent the entire sub-region.

-Your conclusion is not supported by the findings, pls see my comment in the result and discussion section. To higjlight on the contribution of wealth to the favor of the urban women, I should first see which categories of wealth ( poorest, poor, richest etc) dominate where ( urban or rural) and their respective contributions in the decompositions, because wealth is a categorical variable. Also, how do we specifically overcome the differential effect of the determinants to close the urban-rural gap in terms of use of facility delivery? That is an important dimension of your work and is better to further elaborate on how these can be challenged.

-the last statement of the conclusion has an incorrect word “strategist”

6. PLOS authors have the option to publish the peer review history of their article (what does this mean?). If published, this will include your full peer review and any attached files.

Reviewer #1: **Yes: **Almamy M Kante

Reviewer #2: **Yes: **Gebretsadik Shibre

---

## [Author Response · Author response to Decision Letter 0]

9 Apr 2021

Authors’ responses to Reviewers and Editors Comments

Dear Reviewers and editors;

We are very thankful for your invaluable comments. Here under are authors’ responses to the comments raised during review process. 

General comments on Journal Requirements

Comment 1: 1. Please ensure that your manuscript meets PLOS ONE's style requirements, including those for file naming. 

Response 1: As much as possible the authors were read the format and submission guidelines and hopefully it will fit with your recommendation. 

Comment 2: We suggest you thoroughly copyedit your manuscript for language usage, spelling, and grammar. 

Response 2: Melaku Yalew has revised the language part.

Comment 3: d. If you did not receive any funding for this study, please state: “The authors received no specific funding for this work.” Please include your amended statements within your cover letter; we will change the online submission form on your behalf.

Responses 3: The authors received no specific funding for this work

Comment 4: In your Data Availability statement, you have not specified where the minimal data set underlying the results described in your manuscript can be found. PLOS defines a study's minimal data set as the underlying data used to reach the conclusions drawn in the manuscript and any additional data required to replicate the reported study findings in their entirety. All PLOS journals require that the minimal data set be made fully available. For more information about our data policy, please see http://journals.plos.org/plosone/s/data-availability.

Response 4: The data underlying the study can be accessed after legal registration at www.measuredhs.com. After registration, interested researchers can log in at https://www.dhsprogram.com/data/dataset_admin/login_main.cfm and access the data as zipped files.

Comment 5: Your ethics statement should only appear in the Methods section of your manuscript. If your ethics statement is written in any section besides the Methods, please move it to the Methods section and delete it from any other section. Please ensure that your ethics statement is included in your manuscript, as the ethics statement entered into the online submission form will not be published alongside your manuscript.

Response 5: our ethics statement only appears in the Methods section and it is in our manuscript

Comment 6: Please include a copy of Table 5 which you refer to in your text in lines 380 and 707.

Response 6: table 5 is the currently of table 4 

Comment 7: Please include your tables as part of your main manuscript and remove the individual files. Please note that supplementary tables (should remain/ be uploaded) as separate "supporting information" files. 

Response 7: we have included our tables as part of main manuscript and removed the individual files.

Comment 8: Thank you for submitting the above manuscript to PLOS ONE. During our internal evaluation of the manuscript, we found significant text overlap between your submission and the following previously published works

Response 8: we have tried to address this comment by avoiding text overlap from our document as well as citing the citable references for our work.

 Reviewer 1

 Factors affecting the institutional delivery in LMIC is an important topic to explore. The authors used an interesting method to quantify the phenomenon. As authors stated the paper will be the first to use the Oaxaca-Blinder decomposition to examine the urban rural disparity in institutional delivery in selected east African countries

Comment 1: The paper is too long and most of information is not needed here or can be shortened

Response 1: we have tried to concise the document 

Background:

Comment 2: The last paragraph of the background “page 12 lines 112-121”) can be moved to methods section

Response 2: we have deleted it from background

Methods:

Comment 3: Study population is just the women that have delivered in the past 5 years. Question: if a woman had delivered more than once in the past 5 years, please explain how are you exploring the information: are you considering all deliveries or just the last one?

Response 3: all births in the last five years were considered

Results section

 Comment 4: Table 1 needs some statistical tests to these differences..

Table 2 needs some statistical tests and also there is a big problem for percentage of the “pregnancy-related deaths” (0.8%). The PRD is not affecting old women and most of deceases are women 60 years and above. You should recalculate this proportion based only on women of reproductive age (15-49years). This proportion will be much high that what the authors published. I think it will be more than 3%.

Please revise table 1 and table 2. The description in the manuscript is too confusing. I think on page 12, line 261 it is probably Table 2 and line 265 it is Table 1. Please review it again.

Table 1: Why not can provide the percentages instead of numbers? Too confusing.

What is the difference between table 1 and figure 1. I think they are providing the same information. If yes select only one of them

Response 4: Thank you very much, we have included chi-square and t-tests accordingly for categorical variables and continuous variables to show the presence of significant difference in place of residence. We have removed table 2 since the massage is similar with figure 1. Our study participants were 15 to 49 years women not 15 to 60 years and we haven’t any statement “pregnancy-related deaths” (0.8%)” in our document. Please you can check the result. Sorry for creating confusion we have corrected it by deleting table 2. The overlapping information reduced from table 1 and each has its own separate information.

Comment 5 minor comments

Response 5: thank you, we have edited and deleted per comments.

Comment 6: Overall: the paper is too long and I will recommend the authors to shorten it by deleting or reducing several sections. For instance the last paragraph of the background is the same information that was described in the method. The description of the decomposition method is well known and can be shortened and authors can just provide references as needed. The results section can describe briefly the most important results and not provide too much information. At the end the readers are lost. Also the results sections should be shortened by describing results instead of providing some explanations in the results section with references in this section. All explanations and reference in the results section should be moved to Discussion section 

Response 6: for this comments we have deleted parts from background, method and result accordingly without missing the flow of the manuscript. 

Reviewer #2: -

Abstract

Method

Comment 1: -indicate that the study was in four east African countries

Response 2: we have mentioned that our analysis data was from “four East African countries”

Comment 2: covariate effect was dominate”; this contradicts with the result in the body

Response 2: we have corrected the body part that that the covariate effect is the dominant for the change in this study 

Conclusion

Comment 3: -“regional inequality” is confusing;

-“facilitate antenatal care utilization” is not clear, which ANC care? Four or more? Be specific

-“plan number of children” is vague

Response: for clarity we have replaced regional inequality with residential inequalities,

- facilitate antenatal care utilization also replaced with to increase frequency of antenatal care utilization

- Plan number of children removed 

Introduction/background

Comment 4: Introduction/background

-this section is an awkward and poorly written. You incoherently moved between different ideas; you begin with mortality, and then jumped to facility delivery and then back to mortality. Pls rewrite this section and ensure that the transitions between paragraphs are effective and clear. Also, you need to further substantiate it by adding points directly related with urban-rural disparity in facility delivery. Do not forget that you still follow certain chronological flows.

- in the “maternal mortality rate below 70”, change rate into ratio

-Lines 112-120: take this to method section

Response 4: we have critically considered the comment and we have rewritten the background section accordingly, please see the background for confirmation.

-we have changed rate into ratio

-we removed the method section from background 

Methods

Comment 5: DHS involves women 15-49 and men 15 to 59 as well as children under five. You need to mention this in the method section through your analysis in this study has been restricted to only women. You need to also clarify the distinction between “study population” and “sample”. All women in the reproductive age group in the selected households are samples, not study populations, unlike what you have said. Pls avoid the confusion around these terms

Response 5: we have included in the method “DHS involves women 15-49 and men 15 to 59 as well as children under five. For the current study we used women 15-49 years”. 

-Our sample population: - sample population was all reproductive-age women in each household at the enumeration area.

Comment 6: the “sampling and sample size” section is too vague to understand how and how many enumeration areas (ESs) and participants were enrolled into the study. It needs to be clearly expounded that the DHS follows a stratified two-stage cluster design, where, following stratification by both urban and regions/districts (you just mentioned that stratification was based solely on urbanization and that is false), eligible women were selected in two stages. In this section, you can explain that succinctly. How many EAs were sampled in the first stage? How many households were sampled in each EA? What was the household and individual response rates? What is primary sampling unit/cluster and SSU? Explain

Response 6:we have address these comments as “Urban-rural based stratified two stage cluster sampling was employed for each region. In the first stage clusters were selected randomly and in the second stage equal numbers of households were selected using systematic random sampling [3–6]. From Ethiopia, Kenya, Tanzania and Uganda 645, 1612, 608 and 697 clusters were sampled in the first stage respectively. For Ethiopia 28, Kenya 25, Tanzania 22 and Uganda 30 households per cluster were selected in the second stage. From these in Ethiopia 18008, Kenya 40,300, Tanzania 13,360 and Uganda 20,880 households were sampled. Overall the household response rate was 99% for Keya and 98% for Ethiopia, Tanzania and Uganda. From identified eligible women the response rate for women interview was 95% for Ethiopia and 97% for the rest countries. The current study incorporates 643 clusters and 10,547 births from Ethiopia [4], 1,593 clusters and 20,840 births from Kenya [5], 608 clusters and 10,175 births from Tanzania [3] and 696 clusters and 15,154 births from Uganda [6].”

Comment 7: place of residence is the stratifier variable, not an exposure/independent variable. It is a variable the inequality was measured by accounting for the influences of exposure variables

Response 7: we have corrected residence as “Stratification variable” since have learned the issue.

 Comment 8: maternal age (age during the time of the surveys administrations) could not be an exposure variable, because it comes long after the outcome variable occurs. The word “cautious” is ambiguous in the “age at birth” 

Response 8: Thank you “Mr” Gebretsadik Shibre for this critical comment make as to find solutions for the challenge and we compute the exact age of the mother at the birth of their child by subtraction of the year of birth b2 of the Child from the year of birth of the mother (v010). And reanalyze. 

Comment 9: it is strange that you did not explain how these variables were selected, and no citation to prior similar works was made. You need to tell us that these variables show certain theoretical and or statistical relationship with the intended response variables. 

Response 9: we have cited our sources that were used as a base for our classification. 

Comment 10: Region is missing from being an exposure variable

Response 10: the reason we missed regions from exposure variables in our study is that regional classification is not similar to compare across study countries 

Comment 11: Distance from health facility: if this variable was created based on the variable v467d, then that is incorrect. This variable reflects access problem at the time of the survey for a woman's medical help, and is not related with facility delivery.

Response 11: yes we used v467d, because we assume that access is one of the possible hindering factors for institutional delivery.

Comment 12: How was media exposure variable was created and produced? Explain that to allow replication for other studies as well as to improve transparency. This applies also to other variables.

Response 12: for clarity we put in the method section operational definition heading “women are considered as regularly exposed to media if they have at least one of the media (newspaper, radio or television) at least once a week otherwise considered as have no access”. For transparency we generate media exposure from three variables (frequency of reading newspaper or magazine (v157), frequency of listening to radio (v158) and frequency of watching television (v159) and three of them have three responses (not at all (0), less than once a week(1) and at least once a week (2)). Based on this information we generate media exposure as this do file we do

gen media_exposure=.

replace media_exposure=1 if v157==2|v158==2|v159==2

replace media_exposure=0 if media_exposure==.

label variable media_exposure "media exposure"

label define mediaexposure 0 "no accesses media" 1 "have accesses to media"

label values media_exposure mediaexposure 

Comment 13: marital status variable was measured in DHS at the time of the survyes, and might not reflect marital status of the women before becoming pregnant, and therefore might not be an exposure variable for facility delivery; it has to come before or at least as the same time as the outcome variable to call it an exposure variable. Pls see this variable in the DHS guide for scrutiny.

Response 13: Really it is true some variation will be happen at time of birth and at the time of survey since marital status is time varying variable. However, the study design is cross-sectional which measure association but not able establish temporality. For this we acknowledged the limitations of the study. 

Analysis 

Comment 14: how did you assess the normality and Skewness of the variables? Explain that.

Response 14: we applied Skewness and kurtosis test with normal quantile plot to assess normality.

Comment 15: -you have described that the analyses were all ”population-weighted” and that “survey design” was taken into cognizant during analysis. This expression is very misleading for many reasons. First, the phrase “population-weighted” adds noting to understanding except that it brings confusion with it. You simply say you analysis is “weighted”. Second, you did not explain how your analysis should be weighted. Third, when we say, survey design is taken into account, we are saying that we are accounting for the disproportionate sampling (that we weight our analysis) and the stratification and clustering effect are taken into account to produce nationally representative estimates. So, the later expression, that is, the survey design is taken into account is enough and there is no any reason to repeat that analyses were weighted as far as you have mentioned that you have already mentioned that the complex nature of the data has been accounted for. Finally, when you say survey design was accounted for, I am not sure whether you have accounted for both stratification and clustering, and if you have missed one of them, then the estimated Standard errors were most likely be erroneous.

Response 15: we have used only “the survey design is taken into account” 

Comment 16: it is not clear why you have also used the decomposition techniques suggested by powers et al after you have calculated BO decomposition? Both methods can be used of binary variables. But the problem with BO is that results could hugely differ depending on the reference category chosen during analysis, aka “identification problem”. So, with BO, YOU must do normalization to make the finding consistent with the chosen reference categories of categorical variables.

Response 16: so as to get detailed decomposition and normalized decomposition we used powers et al (Yun decomposition) and (nldecompose) for aggregate decomposition. However we missed the dummies and we have included in the revision. 

Comment 17: - in the ethics section, please also mention that ethical procedures were the responsibility of the institutions that carried out the survey.

Response 17: thank you, we have mentioned it.

Results

Comment 18: -Since your outcome is facility delivery, just report that; reporting home deliveries adds only confusion

Response18: we have corrected with facility delivery percentage 

Comment 19: the “delivery rate” is inappropriate usage as rate has a different meaning and does not go with delivery

Response 19: we have removed rates 

Comment 20: the statement “however…, Table-1” is very vague and I cannot understand it

- The statement “Kenya has the highest 249 (35.83%) urban resident women 

Response 20: sorry for making confusion, we have removed it but what we want to express is in the other country far most women reside in rural area.

Comment 21: lines 251-261: the texts here do not correspond to the content of table 1. Presented in table 1 is completely different from what the texts talks about. The texts about the differentials of urban and rural need to be presented well in terms of all the exposure variables so that we can see where these variables dominate, and this would then help us interpret the decomposition findings presented table 2 through table 4.

Response 21: here what we have tried is to compare the study areas by background characteristics with consideration of its importance to our discussion later.

Comment 22: Lines 262-265: it is the replication of table 2; pls avoid repetitions and redundancies

Response 22: really it is repetitions and we have removed 

Comment 23: Lines 287-288: I think the contributions of the variables in the decomposition can be to either widen or narrow down the gap; this is the most interesting portion of the findings and hence requires further explanation in the discussion section. Why some variables contribute in terms of widening the gap, but not others?

Response 23: we did 

Comment 24: lines 290-292: take this into the discussion

Response 24: we have moved it to discussion

Comment 25: -Lines 249-356: this paragraph actually lacks clear interpretations/presentations of findings. When you say that disparity could be reduced if urban and rural settings have similar levels of wealth, for instance, it can also be interpreted that disparity can be lowered if wealth in urban areas is reduced to a level in rural settings, which is not actually the case. So, you need to clearly present that when wealth in rural settings are increased to a level in urban areas, urban-rural disparities can be narrowed by certain amount. Also, make sure that in this text, you have not repeated what you have already presented in the Table. 

Response 25: thank you based on the comment we have analyzed with dummies of the categorical variable to show the clear direction either to wide or narrow the gap. 

Comment 26: Lines 357-359: in this small paragraph, you attempted to attach meaning to the observed contributions of the exposure variables. You are interpreting that variables with positive contribution like wealth tend to narrow down the disparity. However, how do these variables contribute this way? How do I know that these variables narrow, not widen the gap? I rather interpret the opposite way. I strongly recommend you to first document where these variables dominate (in urban or rural areas). For instance, if the richest and richer categories of wealth are predominantly found in the urban area, and if these categories have the capacity to increase the chance of giving birth in facilities, then surely, wealth tends to widen the disparity, rather than narrowing. Also, it is extremely important to show the contributions of each of the five categories of wealth, and of other categorical variables, and without doing that, you cannot interpret whether a variable contributes by widening or narrowing. At currently stands, wealth has 16% contribution to the disparity in Ethiopia, and we cannot know whether this contribution is to close the gap or not. My same comment applies to the other similar findings in the subsequent lines

Response 26: yes we have accepted the comment and addressed it as we have mentioned in response 25. We have also in showed that these factors are significantly different between urban and rural residences. Please see table 1.

Discussion 

Comment 27: Lines 408-410: revise the vague statement

Response 27: we have tried to clarify by shortening sentences and selecting suitable words. 

Comment 28: Lines 414-415: Even though there might not be a study on these countries together, there could be a number of studies that investigated facility delivery inequality in these countries, so unless you are 100% certain that there are no previous studies, you should rather avoid making such bold statement that there were no prio0r studies.

Comment 29: Lines 414-415: Even though there might not be a study on these countries together, there could be a number of studies that investigated facility delivery inequality in these countries, so unless you are 100% certain that there are no previous study, you should rather avoid making such bold statement that there were no prio0r studies.

Response 29: Thank you; yes it needs humble expression that is why we say to “our knowledge”.

Comment 30: Lines 420-427: you need to show that the variation has been explained by the factors whose contributions have been explored and presented in the tables in the result section. For instance, in these lines, you have mentioned that the urban-rural disparity has been partly explained by the disparity in formal education and at least one ANC (or the lack thereof). However, in the result sections, you did not show separately that such variables have contributed to the disparity. Also, the word “ignorance” has nothing to do with your finding and seems also unethical.

Response 30: Yes in the previous document we had missed it but in current document tables 3 and 4 answer this gap since we have reanalyzed per your comment. For the word ignorance we use uneducated mothers. 

Comment 31: lines 433-435: it is important how the unequal distributions of endowments encourage urban women to attend to facility delivery. Which of these variables encourage urban women to deliver at facility, which variables discourage them? Explain the mechanism of actions of these variables to benefit women in urban and rural settings.

Response 31: we have mentioned variables and the possible mechanisms with reference 

Comment 32: -lines 437-440: avoid the repetitions. How these variables widen the urban-rural gap?

Response 32: we have revised it to express additions rather than to repeat what already said 

Comment 33: -lines 446-448: not clear, and needs some clarification to help the statement read well

Response 33: this expression was the combination of (table 1) covariate distribution differences between urban and rural women; and the facility delivery gap the will be narrowed if the gap in this covariate distribution increased in rural women (table 3).

Comment 34: lines 448-450: this interpretation is unclear; the evidence in the result section does not support the interpretations in these lines; how do we know that more frequent ANC visits could help urban women attend facility delivery in the absence of findings on the contributions of the specific categories ( that is 1-3 and >4= ANC visits) of the number of ANC visit?

Response 34: yes we have missed dummies and it was not directional. As we have addressed in response 25, this ambiguity will be solved in the current document. 

Comment 35: for the subsequent paragraphs, there is a need to make a major revision and make sure that every piece of your interpretation is completely informed by your findings. Since the analysis is incomplete, this has seriously affected the interpretations of the findings in this section of the paper.

Response 35: similar with responses 25 and 34

Comment 36: -Also, it is good if you briefly elaborate on the policy implications of the study. 

Response 36: we have tried to elaborate the policy implications for countries need covariate redistribution and behavioral interventions. 

Comment 37: -it is not correct to blame the study being cross sectional as limitations of the study; causality cannot be made by a single study even with RCT, and more importantly, temporality is not the problem in the study at least for some of the variables like education and wealth. One important limitation in this is that in the BO decomposition, findings differ depending on the chosen reference categories of categorical variable, and hence requires further adjustment using “normalization” process. But, you need to overcome this problem in the study to get valid results.

Response 37: thank you for make us reconsider this limitation; however, our insight is that almost 16% of women at birth were less than 20 years. So they can change their education and wealth over time including with marriage or change in marriage. For identification problem we applied normalization. 

Comment 38: I cannot assume that these findings are generalized to the east African community since these four countries do not represent the entire sub-region.

Response 38: we have corrected with this “The finding also generalized for similar countries as the data were representative and quality assured”

Comment 39: Your conclusion is not supported by the findings, pls see my comment in the result and discussion section. To higjlight on the contribution of wealth to the favor of the urban women, I should first see which categories of wealth ( poorest, poor, richest etc) dominate where ( urban or rural) and their respective contributions in the decompositions, because wealth is a categorical variable. Also, how do we specifically overcome the differential effect of the determinants to close the urban-rural gap in terms of use of facility delivery? That is an important dimension of your work and is better to further elaborate on how these can be challenged.

Response 39: we have addressed in result and discussion similar with responses 25 and 34

Comment 40: -the last statement of the conclusion has an incorrect word “strategist”

Response 40: corrected as “strategies”

---

## [Decision Letter · Decision Letter 1]

19 May 2021

PONE-D-20-30414R1

Urban-rural disparities in institutional delivery among women in East Africa: a decomposition analysis

PLOS ONE

Dear Dr. Dewau,

Thank you for submitting your manuscript to PLOS ONE. After careful consideration, we feel that it has merit but does not fully meet PLOS ONE’s publication criteria as it currently stands. Therefore, we invite you to submit a revised version of the manuscript that addresses the points raised during the review process.

We look forward to receiving your revised manuscript.

Kind regards,

Fernando C. Wehrmeister

Academic Editor

PLOS ONE

Additional Editor Comments (if provided):

One reviewer suggested more revisions needed to improve the manuscript. These points should be taken into account by the authors. Please, consider carefully all the comments.

Reviewers' comments:

Reviewer's Responses to Questions

**Comments to the Author**

1. If the authors have adequately addressed your comments raised in a previous round of review and you feel that this manuscript is now acceptable for publication, you may indicate that here to bypass the “Comments to the Author” section, enter your conflict of interest statement in the “Confidential to Editor” section, and submit your "Accept" recommendation.

Reviewer #1: All comments have been addressed

Reviewer #2: (No Response)

2. Is the manuscript technically sound, and do the data support the conclusions?

Reviewer #1: Yes

Reviewer #2: Yes

3. Has the statistical analysis been performed appropriately and rigorously? 

Reviewer #1: Yes

Reviewer #2: I Don't Know

4. Have the authors made all data underlying the findings in their manuscript fully available?

Reviewer #1: Yes

Reviewer #2: Yes

5. Is the manuscript presented in an intelligible fashion and written in standard English?

Reviewer #1: Yes

Reviewer #2: No

6. Review Comments to the Author

Reviewer #1: All comments have been addressed by authors and the paper looks better. Editing was improved and the length of the paper was reduced too.

Reviewer #2: I have provided all my comments and suggestions in a separate file. Authors need to improve the overall readership of the manuscript.

7. PLOS authors have the option to publish the peer review history of their article (what does this mean?). If published, this will include your full peer review and any attached files.

Reviewer #1: **Yes: **Almamy M Kante

Reviewer #2: **Yes: **Gebretsadik Shibre

---

## [Author Response · Author response to Decision Letter 1]

28 May 2021

Author response to Reviewer 

Dear Reviewer, Dr. Gebretsadik Shibrie

We are very grateful for your constructive comments and suggestions. Here under are authors’ responses to the comments raised during second revision. 

Background 

Comment 1: In every piece of your writing in this section, make sure you have touched issues for all the four countries under investigation. But as it currently stands, some paragraphs (e. g. the paragraph that comes after paragraphs which explain about maternal mortality burden) concentrated solely on situations of Ethiopia. Here, you mentioned about the coverage of skilled delivery in Ethiopia as of 2016 EDHS, what about coverage of the same indicator in the other study countries?

Response 1: we have added the figures regarding to the rest three countries. (Kenya 61% [5], in Tanzania 63% [3] and in Uganda 74% [6]) 

Comment 2: In this same paragraph, the statement “……..Even though women have accessed cost free 148 services, the proportion of institutional deliveries in Ethiopia was 26%” is a bit confusing; the physical services are in fact offered free of charge, but they are not available for free as women incur a lot of other costs to get the skilled maternal health care services. I suggest you revise it to read more clearly. The statement in the lines 151-152 is incomplete; it is a phrase.

Response 2: we have rephrase like this, Even though women have accessed cost-free maternal and new-born care services, the proportion of institutional deliveries were in Ethiopia 26%......

Comment 3: The 79% (urban) vs.20% (rural) deliveries do not add up to 100%. Pls revise the figures. 

Response 3: figure 20% is corrected as 21%. 

Comment 4: In this paragraph, the urban-rural scenario of skilled delivery was made for only Ethiopia and Kenya; you need to also do the same for the other countries, which you are going to assess their disparities. Without first establishing whether disparity exists between urban and rural in terms of your response variable, you could not logically go on to explain (through decomposition) the disparity. 

Response 4; for the rest of the countries we have added “In the case of Kenya urban institutional delivery was 82% and 48.5% in rural residents [5], in Tanzania urban 86.4% and rural 53.7% [3] and in Uganda it was reached 87.8% in urban and 69.5% in rural areas [6]. ”

Comment 5: The last statement in this paragraph, that reads “These inconsistencies were 161 also well documented in different countries DHS and WHO reports” does provide only confusion, since no inconsistencies have been mentioned in that paragraph, which inconsistencies you are referring about? Instead, you have discussed similar findings about the existence of urban-rural inequality around institutional deliveries in developing nations including Ethiopia and Kenya.

Response 5: inconsistency was replaced with “These urban-rural discrepancies”

Comment 6: Line 162 begins with “Other explanatory variables”; this has brought a huge confusion because in the preceding paragraph, you just were explaining about the existence of the urban-rural disparity, not about factors that drive the disparity. Mentioning that there is disparity is one thing; describing the factors that could underlie an observed disparity is another thing. I suggest you could give one paragraph for the latter case as the first case has already been done. 

Response 6: Other explanatory variable was replaced with “From different literatures various explanatory variables were significantly linked with facility-based delivery; women education…..”

Comment 7: The statement “However, these studies were lacking to determine 168 the observed differences in rates of health-facility deliveries between rural and urban women” requires more nuanced explanation to make it a strong reason for drawing attention to this topic in the first place; many attempts have already been made to show the difference between urban and rural settings in terms of institutional birth, but probably, what is lacking is that, this difference was not explained rigorously using the decomposition method you are using. 

Response 7: we have rephrase like this, “However, decomposition analysis method help to decomposing inequality in institutional deliveries in Urban-rural settings, this difference was not rigorously explained using the decomposition method.”

Comment 8: I suggest you revise and substantiate this last paragraph of the background as readers want to see where the novelty of this study lies in the midst of a prolific number of similar articles on this same area. Finally, I look forward to seeing a reason that justify the restriction of the present study to just Easter Africa, while it is possible to do the same study in any other settings.

Response 8: we have tried to rephrase as such “To develop effective strategies that minimize this urban-rural gap demands as such, a greater understanding of the underlying factors contributing to these disparities in institutional deliveries between urban and rural has paramount importance. So, this study aimed to identify the sources of variation in institutional delivery between the urban and rural areas in East Africa by using a decomposition analysis method.” Please see the manuscript. 

Methods

Comment 9: The first paragraph that details about the nature and purpose of DHS goes without citation.

Response 9: thank you for reminding we have cited properly. 

Comment 10: The authors have substantially improved the section which details about sampling procedure for each country. But one thing that is conspicuously missing is that, you have said that stratification was done before selection of samples. However, it is not clear whether stratification was done on both place of residence and regions for all the four countries. On line 200, the statement that reads “Urban-rural based stratified two stage cluster sampling was employed for all regions” is too vague to understand; do you mean that stratification was done by region as well as place of residence for Ethiopia? Pls clarify that.

Response 10: it was for all countries urban rural based but for Ethiopia both regional and residence based stratification was applied. 

Comment 11: Further, I am not sure whether the same applies for the other settings. I even wonder whether “regions” are available in the other countries, and other terms might have been used in the place of regions. I suggest to double check this.

Response 11: yes you are right; in other countries regional classifications are geographical and are not similar with Ethiopia. The word regions to mean countries that were included in the study and we have corrected it. 

Comment 12: On the lines 200 and 201, you went on to say that “clusters were selected randomly”; nevertheless, I found this information not true since clusters are normally selected based on Probability Proportional to Size (PPS) approach, not randomly. 

Response 12: really we miss it and we have corrected per your comment and checking the approach from reports. Thank you!

Variables

Comment 13: The way the response variable is measured varies by where deliveries took place; in Ethiopia, health extension workers can be counted as skilled attendant and by extension, delivery in health post can be treated as institutional delivery. So, it is good to briefly summarize how the response variable is measured in all the four settings.

Response 13: For analysis, place of delivery was measured as a binary variable and was classified as “1” if a woman delivered in any healthcare facility and otherwise “0”. any healthcare facility is a facility minimum maternal cares and child care (bleeding control, vaccine and referral service). The name of minimum service providing facilities can be different health post in Ethiopia to dispensary Tanzania. 

Comment 14: I suggest that you change “stratification variable” into equity stratifier. The former confuses with the sampling procedure you mentioned prior to this section.

Response 14: we have changed stratification variable to equity stratifier variable. 

Comment 15: Some explanatory variables require further scrutiny. The variable “occupation” has many categories in DHS and I did not see all of them here. Cite a reason.

Response 15: our classification was for analysis amenity and interpretation. 

Comment 16: How can age of a woman at birth leads to institutional birth? Age at pregnancy could give more sense.

Response 16: yes age at pregnancy is appropriate. However, this age is mathematically computed nearly age at birth minus one year, since there is no recorded pregnancy age for those recent gave births but for currently pregnant women. Moreover the variable is not significant in the current study in both collapsed and as the questionnaire form. 

Comment 17: Why did you collapse wealth quintiles into three categories? Pls justify; shrinking the categories from 5 to 3 could result in loss of important information.

Response 17: Since in different literatures both classifications were used to make our manuscript concise we have reported the three classification approach. 

Comment 18: Any reason for using two methods as the same time; BO and multivariate decomposition (“mvdcmp” command) for decomposition?. I previously commented on the need to clarify whether authors have implemented “normalization” procedure when running BO decomposition as this method suffers from identification problem. Now, in the revised version, authors have provided the following response:

 “so as to get detailed decomposition and normalized decomposition we used powers et al (Yun decomposition) and (nldecompose) for aggregate decomposition”. Here, why then you have used BO decomposition if everything has been done with the former method? Put another way, a BO decomposition that is not normalized results in a finding that varies by the type of reference group you chose in your analysis. For instance, if I were to re-analyze these same data and want to figure out the contribution of wealth, with the richest as referent group, then, surely, I will end up with a finding that is in no way comparable with what you have generated. This is the greater caveat of BO decomposition that could be minimized by undergoing normalization. If you have done this in this revised version, please indicate that. Plus, the “nldecompose” does not support all the three design elements I talked about somewhere in this recommendation.

Response 18: nldecompose with different weighting provide results of Oaxaca, Blinder, Reimer, and Cotton that is why we prefer it for aggregate decomposition.

Authors prefer multivariate decomposition for detailed decomposition analysis and interpretation suit as both methods used in normalized detailed decomposition. As we have read ” mvdcmp is primarily intended for use in nonlinear decomposition and is based on recent contributions, which include convenient methods to handle path dependency (Yun 2004), computing asymptotic standard errors (Yun 2005a), and overcoming the identification problem associated with the choice of a reference category when dummy variables are included among the predictors (Yun 2005b, 2008). 

 To be transparent and for any comments suit if we are expected to learn our command become this one as a sample

mvdcmp residence1, normal (wealth_index1 wealth_index2 wealth_index3| distance1 distance2 | husb_occup1 husb_occup2 husb_occup3 husb_occup4| media_exposure1 media_exposure2| hus_educ1 hus_educ2 hus_educ3| ANCF1 ANCF2 | No_live_child1 No_live_child2 No_live_child3 ): logit p_delivery wealth_index1 wealth_index3 distance1 husb_occup2 husb_occup3 husb_occup4 media_exposure2 hus_educ2 hus_educ3 hus_educ4 ANCF2 No_live_child1 No_live_child3 age [pw=wt]

# In the introduction if the Oaxaca decomposition make confusion we replace with decomposition method. 

Comment 19: On lines 239 and 240, in the statement “These included factors that are known to be associated with 240 institutional delivery and/or area of residency”, you should avoid the “and/or residency” as all of the presumed covariates/explanatory variables should be associated with facility delivery alone, not with place of residence; place of residence in this study serves only one purpose, serving as “equity stratifier”.

Response 19: thank you, but in this comment was addressed based on your previous comment. 

Comment 20: On the wealth index, you could add more information on which household possessions and durable materials were used to construct the index, and whether similar materials were used in all the four study areas. Furthermore, you need to back up the fact that the percentage of each of the four categories of wealth index deviates from 20% as shown in your result. But in the method section, you said that each category of wealth score accounts for 20%. Why this deviation? 

Response 20: this classification is originated from DHS guide. So, all countries use similar classification. “Households are given scores based on the number and kinds of consumer goods they own, ranging from a television to a bicycle or car, and housing characteristics such as source of drinking water, toilet facilities, and flooring materials. These scores are derived using principal component analysis. National wealth quintiles are compiled by assigning the household score to each usual (de jure) household member, ranking each person in the household population by her or his score, and then dividing the distribution into five equal categories, each comprising 20% of the population.” (Uganda,Tanzania , Ethiopia and Kenya DHS) as well as DHS guide. 

As we have mentioned in methods section for the current analysis we merge poorest and poorer to poor and richer and richest to rich and middle as it is so each three category account for 40%, 40% and 20%.

Comment 21: The authors remain stubborn with their reason for ignoring region in their analysis. But, I would argue that, since they analyzed the data for each countries separately, it poses no problem if they include country specific variables in their analysis. Otherwise, omitting an important variable from for one or the other countries amounts to the so-called Omitted Variable Bias (OVB). Similarity of variables is must for pooled analysis, where an author needs to append one data beneath another to create one mega-data.

Response 21: yes it could be another limitation but we have no other reason other than we have mentioned in first review. 

Comment 22: I see a problem in the “distance” variable; since you used this name to proxy access, make sure that the variable “distance” in this study refers exclusively to access. Plus, this variable was gathered at the time of the administration of the surveys, but institutional birth happened five years ago; how can these two variables be related as a presumed factor and outcome? Can this be highlighted as limitation? There are a few more variables that have such similar problems.

Response 22: yes you are right it is one of the limitations of the cross-sectional study design that suffer from temporal relationship. As you said some individuals status during interview may be different from at a time of child birth and it is not for all and we couldn’t avoid in cross-sectional study. Moreover, if see years lived in place of residence (v104) more than 90% were resided 5 and more years and as you Know the expansion of health facility is not significantly high in this period. It was mentioned as a limitation in the first manuscript. 

Analysis

Comment 23: It is not clear whether you have taken into account of all the three design elements (weighting, stratification, and clustering). It is extremely important for authors to be honest and explicitly state their analysis for other researchers to be able to replicate the analysis. DHS manual highly encourages researchers who are relying on DHS data to account their analysis for all of the three design elements. 

Response 23: to be honest the only think we account is probability weighting. The reason is we cannot found a command that can account the three survey natures. We have tried to see the effect of 

svyset v021 [pweight=wt], strata(v022) vce(linearized) singleunit(missing) on mvdcmp but it has no effect. But only weight has shown difference from unweight outcome. 

Result

Comment 24 I found it very difficult to judge whether a contribution is to narrowing or widening the urban-rural gap. I think it is worthwhile to briefly discuss in the method section on the techniques used to interpret the positive and negative contributions. For instance, how does “poor” contributed to widen the gap? since the largest proportion of poor was found in rural settings, you may be tempted to arguing that it widens the gap, but the same logic applies for rich category as the largest portion of this category is found among the urban settings. So, I may conclude that the rich as well widens the gap, but this time by helping women in urban areas to get most of the institutional delivery. Here, the difference between poor and rich’s contribution is the mechanism of action; the poor by preventing rural women from getting the service and the rich by promoting the urban women to get the service. So, I strongly recommend you to clearly state how one can easily interpret how and why an exposure variable contributes to widen or shrunken the observed urban-rural disparity. To this end, it would be useful to fit a logistic regression for urban and rural settings separately to see whether exposure variables have positive or negative effect on your outcome variable, and this would greatly assist the interpretations of their influence on the disparity. If the effect of the poor group is to really prevent women from getting their child delivered at facility, and if the largest percentage of the poor concentrates among the rural settings, then the poor could widen the gap. If the effect of the rich is to increase the uptake of institutional delivery and if the largest percentage of the rich is found among the urban settings, then, again, this widens the gap by stimulating women to get the service more than the poor rural women do. But, we do not this hidden interesting mechanism of action until you run a logistic regression to see the effect on the outcome variable of the exposure variables. Then, use the regression result to interpret the contributions as well as the mechanisms of action of the variables. Just take a look at the following statement from your analysis and see how unclear it is for readers, On line 484-485:

“poor wealth status (-36.55%) contributed for 485 widening of the gap and rich wealth status (70.40%) contributed for narrowing of this gap.” Why does the poor widen? Why the rich does helps to shrink the gap? See my explanation above to get this problem fixed.

Response 24: for the fear of the length of the manuscript and the reviewer 1 comment most of the methodology should be cited not fully discussed. We prefer to only cite the materials. To clarify our result, the negative coefficients indicate the less sensitivity of the higher group for that particular factor like poor wealth that means if rural poor women became as equal as urban women wealth the institutional delivery in rural women become higher than urban women. The protective effects of poor wealth in home delivery are not as strong for urban as they are for rural. We have tried to discuss the meaning and implication of the wealth in the discussion part and we showed urban rural significant wealth difference in the descriptive analysis. 

The implication of positive coefficient is if that factor become equalised between lower group and higher group (rural and urban) the outcome (institutional delivery difference in high and lower group become narrow). The implication of negative coefficient is that the higher groups are less sensitive to that factor and if the lower groups get the chance of equality with their factor of difference the change in outcome may become greater than the higher group. 

Regarding to logistic regression we have fitted it before we conduct multivariate decomposition to select the factors and visualise their effect. Why we miss in the result is to minimise size of the manuscript. For a simple demonstration the effect of wealth in urban and rural separately supports our argument. 

Comment 25: Finally, what does unexplained portion of the decomposition imply? Does this correspond to discrimination? Pls explain it as it has a huge policy implication.

Response 25: Unexplained (coefficient) effect is the effect of the factor rather than the distribution of factors or behavioural changes and that need quality of the factors rather than the distribution of the factors. We have put the policy implication of unexplained portion immediately before the limitation. 

Limitations

Comment 26: The problems associated with the use of BO decomposition has not been duly explained in this section. The sheer use of BO decomposition is neither a problem nor a limitation by itself; pls describe a specific attribute of the method that you can consider as a weakness or limitation.

Response 26: Although Oaxaca method has limitations like, Ordering of variables problem, Index problem, Observations matching problem and Choice of sample weights, we have applied Oaxaca extension mvdcmp approach to hand this. That is why preferred to reserve from mentioning limitations. 

Comment 27: You have described that temporality remains to be a problem for the study. However, this limitation was not taken into account while interpreting your findings; if the temporality of the exposure variables were not known, why have you interpreted the findings as if they are like causes for institutional birth? When you say that wealth contributes to facility delivery, you are labelling at as a determinant factor. Pls pay attention to this discrepancy.

Response 27: The authors consider, understanding the cross-sectional nature of the study and appreciating its limitation like temporality problem in the limitation section will portray this gaps. 

Comment 28: I highly recommend them to get the paper copyedited by someone highly proficient in English as poorly written manuscript is likely not to provide messages it intends to portray to readers. Also, I highly encourage them to edit the entire paper for some technical additions.

Response 28: we have tried to show our paper for language and grammar editing with language expert in our university 

Comment 29: Finally, I would like to thank the authors for their great work and for substantially improving the first version.

Response 29: Really I am speechless to thank reviewer (Gebretsadik Shibrie) for your time dedication, constructive comments and helping in extensive improvement of the manuscript. Given comments are not only for current manuscript improvement but also increasing our overall understanding of the analysis and research. We may not satisfy you in addressing all issues even we have tried, but we really learnt from what we have done in future as well.

---

## [Editor Report · Decision Letter 2]

12 Jul 2021

Urban-rural disparities in institutional delivery among women in East Africa: a decomposition analysis

PONE-D-20-30414R2

Dear Dr. Dewau,

We’re pleased to inform you that your manuscript has been judged scientifically suitable for publication and will be formally accepted for publication once it meets all outstanding technical requirements.

Kind regards,

Fernando C. Wehrmeister

Academic Editor

PLOS ONE
---

## [Editor Report · Acceptance letter]

22 Jul 2021

PONE-D-20-30414R2 

Urban-rural disparities in institutional delivery among women in East Africa: a decomposition analysis 

Dear Dr. Dewau:

I'm pleased to inform you that your manuscript has been deemed suitable for publication in PLOS ONE. Congratulations! Your manuscript is now with our production department. 

Kind regards, 

on behalf of

Dr. Fernando C. Wehrmeister 

Academic Editor

PLOS ONE